# Ventral pallidum neurons dynamically signal relative threat

Mahsa Moaddab [1✉], Madelyn H. Ray [1] & Michael A. McDannald [1✉]

The ventral pallidum (VP) is anatomically poised to contribute to threat behavior. Recent studies report a VP population that scales firing increases to reward but decreases firing to aversive cues. Here, we tested whether firing decreases in VP neurons serve as a neural signal for relative threat. Single-unit activity was recorded while male rats discriminated cues predicting unique foot shock probabilities. Rats' behavior and VP single-unit firing discriminated danger, uncertainty, and safety cues. Two populations of VP neurons dynamically signaled relative threat, decreasing firing according to foot shock probability during early cue presentation, but disproportionately decreasing firing to uncertain threat as foot shock drew near. One relative threat population increased firing to reward, consistent with a bi-directional signal for general value. The second population was unresponsive to reward, revealing a specific signal for relative threat. The results reinforce anatomy to reveal the VP as a neural source of a dynamic, relative threat signal.

[1] Department of Psychology and Neuroscience, Boston College, 140 Commonwealth Avenue 514 McGuinn Hall, Chestnut Hill, MA, USA.
✉email: moaddab@bc.edu; michael.mcdannald@bc.edu

Environmental threats lie on a continuum from danger to safety, with most threats involving uncertainty. Determining relative threat—where present threat lies on the continuum—allows for an adaptive fear response. Brain regions essential to fear, most notably the central amygdala[1–3] and basolateral amygdala (BLA)[4–8], must be necessary to signal and utilize relative threat. At the same time, threat learning and behavior is the product of a larger neural network[9,10] that includes brain regions traditionally implicated in reward[11–15]. The ventral pallidum (VP) is a compelling candidate for a neural source of relative threat. Anatomically, the VP is positioned to send and receive threat information. So, although best known as an output of the mesolimbic system[16,17], the VP receives direct projections from the central amygdala[18–21] and projects directly to the BLA[22,23].

The VP is consistently implicated in reward processes[21,24–30]. VP neurons acquire firing to reward-predictive cues[31–33] and show differential firing to cues predicting different reward sizes[34,35]. VP neurons change their firing when a taste changes from palatable to aversive[36,37]. More recent work suggests that VP neurons signal relative reward value. Single VP neurons track palatability in a multi-reward setting, showing firing increases that scale with palatability[38]. Yet, VP neurons do not exclusively signal relative reward value. The VP contributes to the formation of a conditioned taste aversion[21,22,39–41] and VP neurons can acquire responding to aversive cues[42].

Whereas earlier work examined the activity and function of VP neurons indiscriminately, more recent work ties function to neurochemical identity[19–21,43]. Most pertinent, mice VP GABA neurons show firing increases to reward cues, but firing decreases to aversive cues[19]. VP neurons showing firing increases to a reward cue and decreases to an aversive cue have also been observed in monkeys[44]. Consistent across both studies, the VP contained a separate population that showed firing increases to both reward and aversive cues, indicative of salience signaling[19,44–46]. Salience signaling has been most strongly linked to VP glutamate neurons[45].

Here, we test the hypothesis that VP neurons signal relative threat through firing decreases. We recorded VP single-unit activity from male rats undergoing fear discrimination consisting of cues predicting unique foot shock probabilities: danger ($p = 1.00$), uncertainty ($p = 0.25$), and safety ($p = 0.00$). Using foot shock outcome permitted direct examination of threat, as shock-predictive cues produce species specific defensive behavior[47,48]. Fear discrimination took place over a baseline of reward seeking[49] and complete discrimination was observed. The behavior/recording approach allowed us to reveal activity patterns reflecting relative threat, relative value spanning threat and reward through opposing changes in firing, as well as salience.

## Results

Male, Long Evans rats ($n = 14$) were moderately food-deprived and trained to nose poke in a central port to receive a reward (food pellet). Nose poking was reinforced throughout fear discrimination, but poke-reward and cue-shock contingencies were independent. During fear discrimination, three distinct auditory cues predicted unique foot shock probabilities: danger ($p = 1.00$), uncertainty ($p = 0.25$), and safety ($p = 0.00$) (Fig. 1a). Each fear discrimination session consisted of 16 trials: 4 danger, 2 uncertainty shock, 6 uncertainty omission, and 4 safety, mean 3.5 min inter-trial interval. Each trial started with a 20 s baseline period followed by 10 s cue presentation. Foot shock (0.5 mA, 0.5 s) was administered 2 s following cue offset on shock trials (Fig. 1b). Trial order was randomized for each rat, each session. Fear was measured by the suppression of rewarded nose poking. A

suppression ratio was calculated by comparing nose poke rates during baseline and cue periods (see "Methods" section for details). Suppression of rewarded nose poking was used because it is an objective, continuous measure of fear output[50]. Nose poke suppression provided a precise trial-by-trial measure of fear output, which was required for regression analyses. After eight discrimination sessions, rats were implanted with drivable microelectrode bundles dorsal to the VP (Fig. 1c). Following recovery, VP single-unit activity was recorded while rats underwent fear discrimination. The microelectrode bundle was advanced through the VP in ~84 μm steps every other day.

Electrode placement was confirmed with immunohistochemistry for substance P[22] (Fig. 1d). Only placements below the anterior commissure and within the dense substance P field were accepted (Fig. 1e, see "Methods" section for details). A total of 435 VP neurons were recorded from 14 rats over 194 sessions. To identify cue-responsive neurons in an unbiased manner, we compared mean firing rate (Hz) during the 10 s prior to cue presentation (baseline), to mean firing rate (Hz) during the first 1 s and last 5 s of cue presentation. A neuron was considered cue-responsive if it showed a significant change in firing from baseline (increase or decrease; paired, two-tailed t-test, $p < 0.05$) to danger, uncertainty or safety during either the first 1 s or the last 5 s interval. This screen identified 257 cue-responsive neurons (~59% of all recorded neurons) from 153 sessions, with at least one cue-responsive neuron identified in each of the 14 rats. All remaining analyses focused on cue-responsive neurons ($n = 257$) and the discrimination sessions ($n = 153$) in which they were recorded.

Rats showed complete discrimination during sessions from which cue-responsive neurons were recorded (mean individual suppression ratio data shown in Supplementary Fig. 1 and session by session individual suppression ratio data shown in Supplementary Fig. 2). Suppression ratios were high to danger, intermediate to uncertainty, and low to safety (Fig.1f). Analysis of variance (ANOVA) for mean individual suppression ratio [factor: cue (danger, uncertainty, and safety)] revealed a main effect of cue ($F_{2,26} = 75.34$, $p = 1.52 \times 10^{-11}$, partial eta squared ($\eta_p^2$) = 0.85, observed power (op) = 1.00). Differential suppression ratios were observed for each cue pair. The 95% bootstrap confidence interval for differential suppression ratio did not contain zero for danger vs. uncertainty (mean = 0.28, 95% CI [(lower bound) 0.19, (upper bound) 0.38]), uncertainty vs. safety (M = 0.52, 95% CI [0.35, 0.65]), and danger vs. safety (M = 0.80, 95% CI [0.65, 0.98]; Fig. 1f). Observing complete fear discrimination permits a rigorous examination of VP threat-related firing.

**Diversity in VP baseline firing and threat responding.** Plotting baseline firing rate, cue and reward firing for each cue-responsive neuron revealed diversity of patterned firing with three prominent features: (1) a mixture of cue-excited and cue-inhibited neurons, (2) showing greatest firing changes to danger, and (3) marked variation in baseline firing rate (Fig. 2). To reveal functional VP neuron types, we averaged the first 1 s and last 5 s danger firing rate for each neuron to obtain a single value and compared this value to zero. Neurons with positive values (>0) increased danger firing rate over baseline and were designated as cue-excited ($n = 131$, ~51% of all cue-responsive neurons; Fig. 2 top). Neurons with negative values (<0) decreased firing rate below baseline and were designated as cue-inhibited ($n = 126$, ~49% of all cue-responsive neurons; Fig. 2 bottom). For cue-excited neurons, we used analysis of covariance (ANCOVA) to determine if baseline firing rate—a candidate marker for neuron type[51]—informed the cue firing pattern. ANCOVA [covariate: baseline firing rate; within factors: cue (danger, uncertainty, and safety) and interval (250 ms bins 2 s prior to cue onset → 2 s

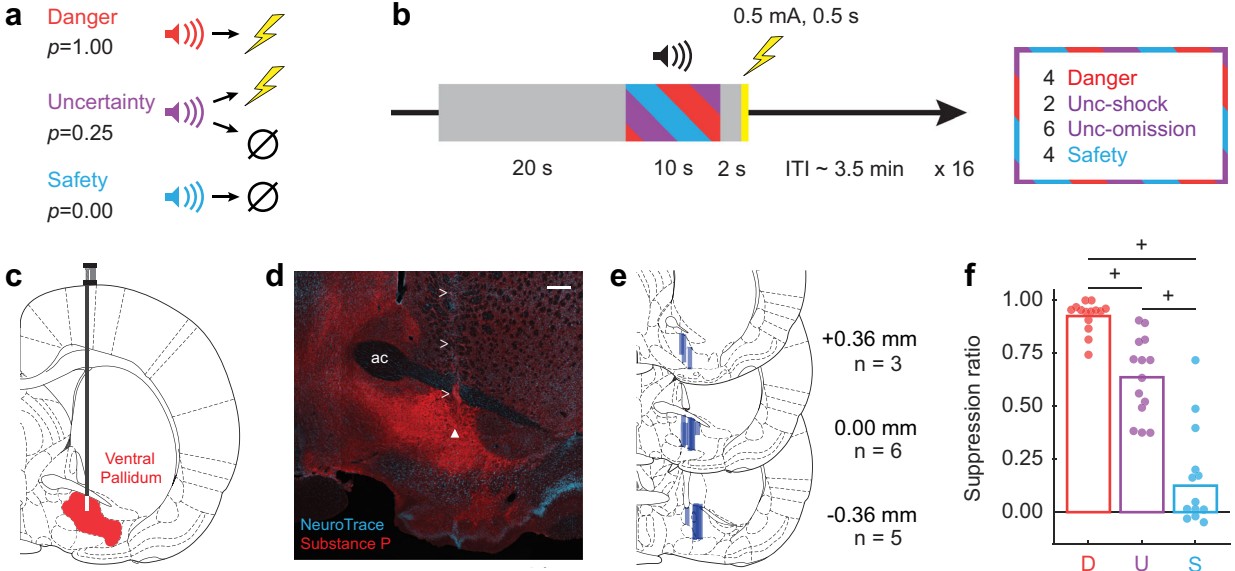

**Fig. 1 Fear discrimination, histology, and behavior. a** Pavlovian fear discrimination consisted of three auditory cues, each associated with a unique probability of foot shock: danger ($p = 1.00$, red), uncertainty ($p = 0.25$, purple), and safety ($p = 0.00$, blue). **b** Each trial started with a 20 s baseline period followed by 10 s cue period. Foot shock (0.5 mA, 0.5 s) was administered 2 s following the cue offset in shock and uncertainty shock trials. Each session consisted of 16 trials: four danger trials, two uncertainty shock trials, six uncertainty omission trials and four safety trials with an average inter-trial interval (ITI) of 3.5 min. **c** After training, a drivable 16 microelectrode bundle was implanted dorsal to the VP. **d** Example of substance P immunohistochemistry (red) showing the recording track (marked by greater than signs) and the location of the recording site (marked by the white triangle) within the boundaries of the VP (NeuroTrace in blue). Scale bar = 50 μm; ac, anterior commissure. **e** Histological reconstruction of microelectrode bundle placements ($n = 14$) in the VP are represented by dark blue bars, bregma levels indicated. **f** Mean (bar) and individual (data points) suppression ratio for each cue (D, danger, red; U, uncertainty, purple; S, safety, blue) is shown for all recording sessions with cue-responsive neurons for all rats ($n = 14$). +95% bootstrap confidence interval for differential suppression ratio does not contain zero.

following cue offset)] found no baseline × cue × interval interaction ($F_{110,14080} = 1.12$, $p = 0.18$, $\eta_p^2 = 0.009$, op $= 1.00$). Because baseline firing rate did not inform the cue firing pattern for cue-excited neurons, all remaining analyses treated cue-excited neurons as a single population.

We separately applied ANCOVA to cue-inhibited neurons to determine if baseline firing rate informed the cue firing pattern. Now, ANCOVA revealed a significant baseline × cue × interval interaction ($F_{110,13420} = 1.93$, $p = 2.34 \times 10^{-8}$, $\eta_p^2 = 0.02$, op $= 1.00$). To identify distinct functional neuron types, we used k-means clustering for baseline firing rate and four additional characteristics: coefficient of variance[52,53], coefficient of skewness[53], waveform half duration[54], and waveform amplitude ratio[54] (see "Methods" section for full description of each). ANOVA revealed four of the five characteristics significantly contributed to clustering, with baseline firing rate being the greatest contributor (baseline firing rate, $F_{2,123} = 546.73$, $p = 6.25 \times 10^{-62}$; coefficient of variance, $F_{2,123} = 8.79$, $p = 0.0003$; coefficient of skewness, $F_{2,123} = 18.20$, $p = 1.20 \times 10^{-7}$; waveform half duration, $F_{2,123} = 17.90$, $p = 1.50 \times 10^{-7}$; and waveform amplitude ratio, $F_{2,123} = 2.12$, $p = 0.12$; firing and waveform characteristics can be found in Supplementary Fig. 3). As a result, cue-inhibited neurons could be divided into three clusters that differed primarily in baseline firing rate: Low firing ($n = 74$), Intermediate firing ($n = 34$), and High firing ($n = 18$) neurons. Between-cluster differences in patterned cue firing were confirmed by ANOVA returning a significant cluster × cue × interval interaction for all comparisons (Low vs. Intermediate, Low vs. High, and Intermediate vs. High; all $F > 1.40$, all $p < 0.005$).

Low firing neurons were observed in 11 of 14 rats and Intermediate firing neurons in 9 of 14 rats, making these neurons likely to be representative of the VP. High firing neurons were observed in only 5 of 14 rats, with 11 of 18 High firing neurons

coming from a single rat (PA02, Supplementary Fig. 1). Because we cannot be certain High firing neurons are representative of the VP, primary analyses focus on Low and Intermediate firing neurons. High firing neuron analyses are provided as supplements (Supplementary Figs. 4, 5, 6a, d, and 7c).

**Differential inhibition of firing is maximal to danger**. If VP cue-inhibited neurons signal relative threat through firing decreases, greatest firing inhibition should be observed to danger, the cue associated with the highest foot shock probability. Lesser and more similar firing inhibition should be observed to uncertainty and safety; whose foot shock probabilities are closer to one another. To determine if differential cue firing was observed, we separately performed ANOVA for Low and Intermediate firing neurons [factors: cue (danger, uncertainty, and safety) and interval (250 ms bins from 2 s prior to cue onset → 2 s following cue offset)]. The cue response pattern for Low firing neurons complied with requirements of a neural signal for relative threat. Low firing neurons showed greatest inhibition of firing to danger, modest inhibition to uncertainty and no inhibition to safety. The relative firing pattern was maintained throughout cue presentation (Fig. 3a). Confirming differential firing, ANOVA for normalized firing rate (Z-score) for the Low firing neurons revealed a significant main effect of cue ($F_{2,142} = 23.45$, $p = 1.58 \times 10^{-9}$, $\eta_p^2 = 0.25$, op $= 1.00$), interval ($F_{55,3905} = 4.66$, $p = 1.18 \times 10^{-26}$, $\eta_p^2 = 0.06$, op $= 1.00$), and most critically a significant cue × interval interaction ($F_{110,7810} = 3.27$, $p = 1.80 \times 10^{-27}$, $\eta_p^2 = 0.04$, op $= 1.00$).

To determine if population-level firing patterns were observed in single units, we constructed 95% boot strap confidence intervals for normalized firing rate for each cue (compared to zero), as well as for differential firing: (danger vs. uncertainty) and

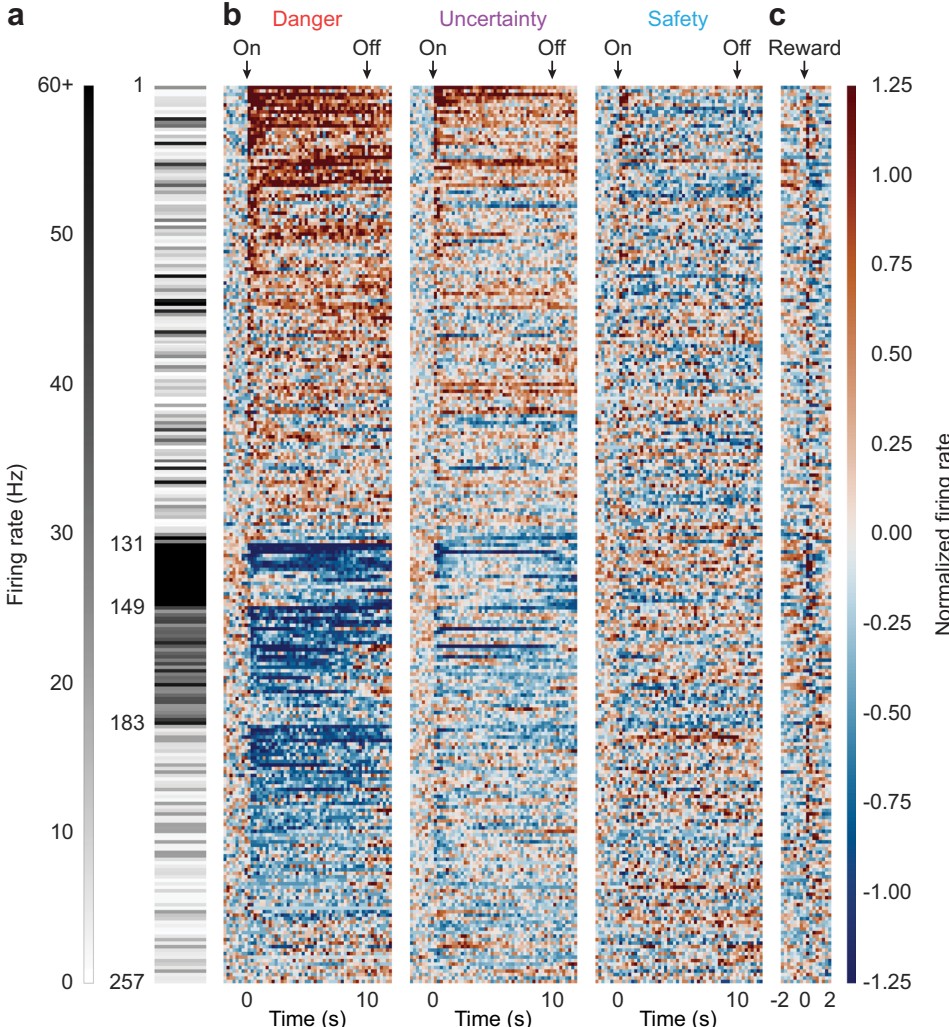

**Fig. 2 Heat plot of cue-responsive neurons. a** Heat plot showing mean baseline firing rate (10 s prior to cue onset) for each cue-responsive neuron ($n =$ 257). Color scale for baseline firing rate is shown to the left, white indicates low baseline firing rate and black high baseline firing rate. **b** Mean normalized firing rate for each cue-responsive neuron, from 2 s prior to cue onset to 2 s following cue offset, in 250 ms bins for each of the three trial types: danger, uncertainty, and safety. Cue onset (On) and offset (Off) are indicated by black arrows. All cue-responsive neurons are sorted by the direction of their response to danger cue (cue-excited, $n = 131$, maroon, top; cue-inhibited, $n = 126$, dark blue, bottom). Color scale for normalized firing rate is shown to the right. A normalized firing rate of zero is indicated by the color white, with greatest increases maroon and greatest decreases dark blue. **c** Mean normalized firing rate for each cue-responsive neuron from 2 s prior to 2 s following reward delivery, advancement of pellet dispenser (colors maintained from **b**). Reward delivery is indicated by black arrow.

(uncertainty vs. safety). Separate 95% bootstrap confidence intervals were constructed for cue onset (first 1 s cue interval), late cue (last 5 s cue interval), and delay (2 s following cue offset) periods. Observing 95% bootstrap confidence intervals that do not contain zero supports interpretations of a firing departure from baseline and differential cue firing.

Low firing neurons showed selective firing inhibition to threat cues at onset (danger: $M = -0.37$, 95% CI [$-0.44$, $-0.29$]; uncertainty: $M = -0.14$, 95% CI [$-0.22$, $-0.06$]) and during late cue (danger: $M = -0.27$, 95% CI [$-0.34$, $-0.19$]; uncertainty: $M = -0.09$, 95% CI [$-0.15$, $-0.04$]; Fig. 3b). Low firing neurons showed differential firing to danger and uncertainty at onset ($M = -0.23$, 95% CI [$-0.31$, $-0.13$]; Fig. 3b, left) and during late cue ($M = -0.17$, 95% CI [$-0.25$, $-0.09$]; Fig. 3b, middle). Similar firing was observed to uncertainty and safety at onset ($M = -0.08$, 95% CI [$-0.19$, $0.02$]; Fig. 3b, left), but differential firing was observed during late cue ($M = -0.14$, 95% CI [$-0.25$, $-0.04$]; Fig. 3b, middle) and delay ($M = -0.12$, 95% CI [$-0.26$, $-6.32 \times 10^{-4}$]; Fig. 3b, right).

Intermediate firing neurons also showed a firing pattern consistent with a neural signal for relative threat. At cue onset, there was greatest firing inhibition to danger, lesser inhibition to uncertainty and least inhibition to safety. Firing inhibition that was specific to danger and uncertainty was maintained for the remainder of cue presentation (Fig. 3c). In support, ANOVA revealed a significant main effect of cue ($F_{2,66} = 27.25$, $p = 2.36 \times 10^{-9}$, $\eta_p^2 = 0.45$, op $= 1.00$), interval ($F_{55,1815} = 7.62$, $p = 1.52 \times 10^{-50}$, $\eta_p^2 = 0.19$, op $= 1.00$), and a significant cue × interval interaction ($F_{110,3630} = 3.90$, $p = 2.20 \times 10^{-36}$, $\eta_p^2 = 0.11$, op $= 1.00$). Single-unit analyses confirmed the ANOVA results. Intermediate firing neurons were inhibited to all cues at onset (danger: $M = -0.54$, 95% CI [$-0.69$, $-0.39$]; uncertainty: $M = -0.25$, 95% CI [$-0.40$, $-0.08$]; safety: $M = -0.21$, 95% CI [$-0.30$, $-0.11$]), but were selectively inhibited to danger and uncertainty during late cue (danger: $M = -0.49$, 95% CI [$-0.60$, $-0.36$]; uncertainty: $M = -0.23$, 95% CI [$-0.34$, $-0.12$]; Fig. 3d). Differential inhibition of firing to danger and uncertainty was observed at cue onset ($M = -0.29$, 95% CI [$-0.44$, $-0.14$];

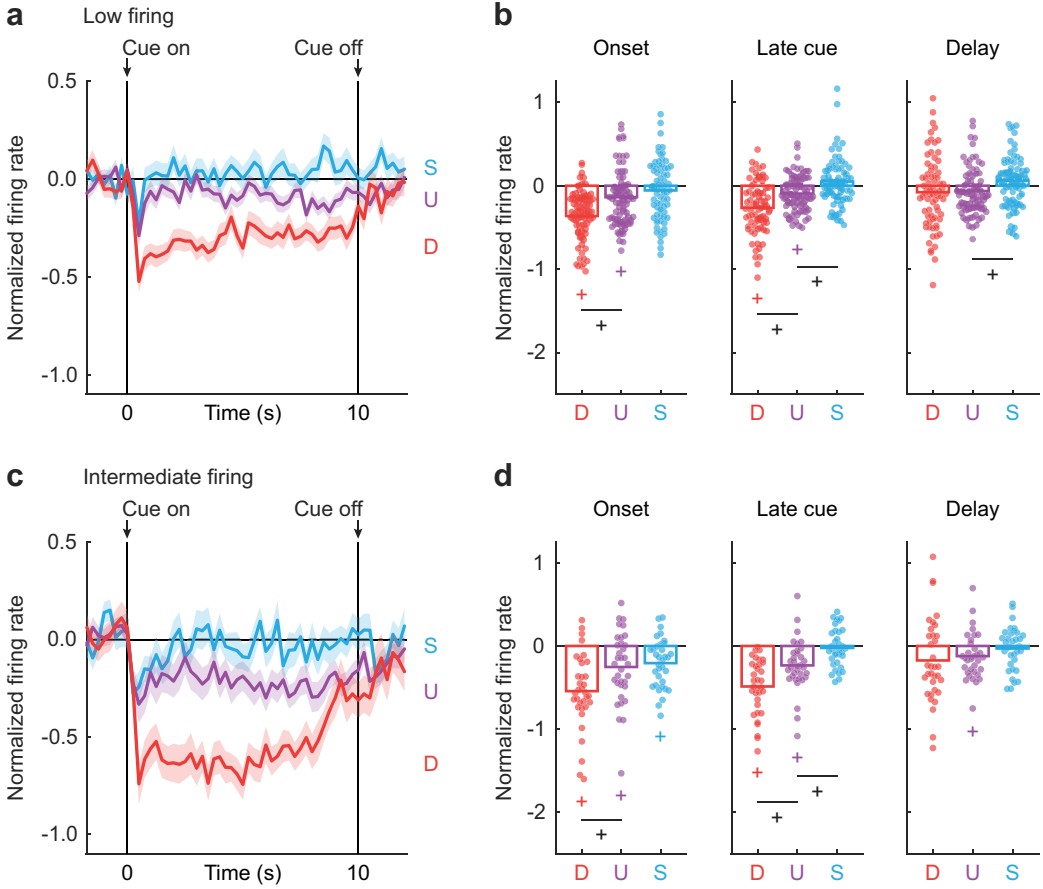

**Fig. 3 Low firing and Intermediate firing neurons preferentially decrease firing to threat cues. a** Mean ± SEM normalized firing rate to danger (D, red), uncertainty (U, purple) and safety (S, blue) is shown from 2 s prior to cue onset to 2 s following cue offset for the Low firing neurons ($n = 74$). Cue onset and offset are indicated by vertical black lines. SEM is indicated by shading. **b** Mean (bar) and individual (data points), normalized firing rate for Low firing neurons during the first 1 s cue interval (onset, left), the last 5 s cue interval (late cue, middle), and 2 s following cue offset (delay, right) are shown for each cue (D, danger, red; U, uncertainty, purple; and S, safety, blue). **c** Mean ± SEM normalized firing rate for the Intermediate firing neurons ($n = 34$), shown as in **a**. **d** Mean (bar) and individual (data points), normalized firing rate for Intermediate firing neurons, as in **b**. +95% bootstrap confidence interval for differential cue firing does not contain zero. +95% bootstrap confidence interval for normalized firing rate does not contain zero (colored plus signs).

Fig. 3d, left), and during late cue ($M = -0.26$, 95% CI [−0.42, −0.10]; Fig. 3d, middle). Like for Low firing neurons, differential inhibition of firing was not observed to uncertainty and safety at cue onset ($M = -0.05$, 95% CI [−0.23, 0.14]; Fig. 3d, left), but was observed during late cue ($M = -0.21$, 95% CI [−0.34, −0.05]; Fig. 3d, middle).

Population and single-unit firing analyses reveal Low and Intermediate firing neurons are candidate sources of relative threat signaling. Even more, positive firing relationships were commonly observed for threat cues, danger and uncertainty, but zero or even negative firing relationships were observed for uncertainty and safety (Supplementary Fig. 5). Firing inhibition was not simply due to the cessation of nose poking. Pauses in nose poking in the absence of cues during the inter-trial interval were insufficient to inhibit the activity of Low and Intermediate firing neurons (Supplementary Fig. 6). Of course, differential cue firing would also be expected of a neural signal for fear output. Given that our rats showed complete behavioral discrimination of danger, uncertainty, and safety; VP firing decreases could reflect fear output, rather than relative threat.

**Low and Intermediate firing neurons dynamically signal relative threat.** We used linear regression to determine the degree to which VP single-unit activity reflected fear output and relative threat (see "Methods" section). Fear output and relative threat could be dissociated because rats showed higher suppression ratios to uncertainty than would be expected based on its foot shock probability (Fig. 1f). For each single unit, we calculated the normalized firing rate for each trial (16 total: 4 danger, 8 uncertainty, and 4 safety trials) for a total of 14 s (1 s bins; 2 s prior to cue onset, 10 s cue presentation, and 2 s following cue offset). The fear output regressor was the cue suppression ratio for that specific trial. The relative threat regressor was a numerical value assigned to each cue. Although known to the experimenters, the rats and their VP neurons had no a priori knowledge of the foot shock probability assigned to uncertainty (0.25). It is then possible that firing decreases reflected relative threat, but were best captured by an alternative probability. To examine this possibility, the values assigned to danger (1.00) and safety (0.00) were fixed, but the value assigned to uncertainty was incremented from 0 to 1 in 0.25 steps (0.00, 0.25, 0.50, 0.75, and 1.00). Regression was separately performed for each of the five uncertainty assignments. Regression output was a beta coefficient, quantifying the strength (greater distance from 0 = stronger) and direction (>0 = positive and <0 = negative) of the predictive relationship between each regressor and single-unit firing (see Supplementary Fig. 7 for full regression results).

A threat tuning curve was constructed by averaging beta coefficients across the 10 s cue for both regressors at each of the

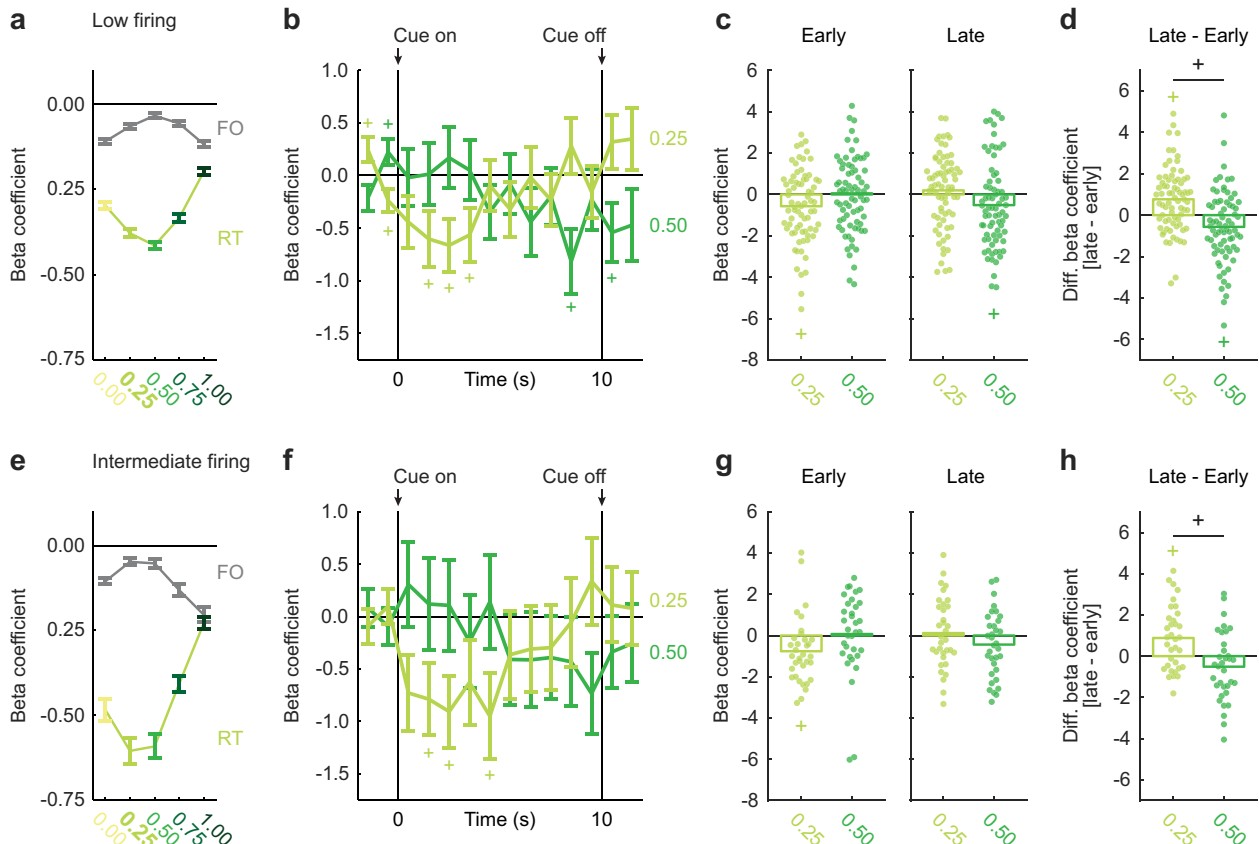

**Fig. 4 Low firing and Intermediate firing neurons dynamically signal relative threat. a** Mean ± SEM beta coefficients are shown for each regressor (RT, relative threat; FO, fear output), during the 10 s cue presentation, for each assignment from 0 to 1 in 0.25 increments (0.00, 0.25, 0.50, 0.75, and 1.00), for the Low firing neurons (n = 74). **b** Mean ± SEM beta coefficients are shown for two relative threat regressors (0.25, assignment, green yellow; 0.50, assignment, green), from 2 s prior to cue onset to 2 s following cue offset in 1 s intervals, for the Low firing neurons. Cue onset and offset are indicated by vertical black lines. **c** Mean (bar) and individual (data points), early (first 4 s of cue) and late (last 2 s cue plus 2 s delay) beta coefficients for two relative threat regressors (assignment, 0.25 and 0.50) are shown for Low firing neurons (colors maintained from **b**). **d** Mean (bar) and individual (data points), differential beta coefficients ([late - early]) for two relative threat regressors (assignment, 0.25 and 0.50) are shown for Low firing neurons. **e–h** Identical graphs made for the Intermediate firing neurons (n = 34), as in **a–d**. +95% bootstrap confidence interval for differential beta coefficient does not contain zero. +95% bootstrap confidence interval for beta coefficient does not contain zero (colored plus signs).

five uncertainty assignments. Low firing neurons more strongly signaled relative threat, compared to fear output, across all five assignments (Fig. 4a). Relative threat signaling was better captured by the actual shock probability (assignment = 0.25), compared to an assignment that equated uncertainty to safety (assignment = 0.00). However, relative threat signaling of the actual probability was similar to signaling of the greater-than-actual, midpoint probability (assignment = 0.50). In support, ANOVA for beta coefficients [factors: assignment (0.00 and 0.25), regressor (fear output and relative threat), and interval (1 s bins from 2 s prior to cue onset → 2 s following cue offset)] revealed an assignment × regressor × interval interaction ($F_{13,936} = 2.13$, $p = 0.011$, $\eta_p^2 = 0.03$, op = 0.96). By contrast, ANOVA for 0.25 and 0.50 beta coefficients found no assignment × regressor × interval interaction ($F_{13,936} = 0.88$, $p = 0.57$, $\eta_p^2 = 0.01$, op = 0.55).

We were curious whether relative threat signaling of the actual probability was indistinguishable from the midpoint probability, or whether signaling dynamically changed as foot shock drew near. Now, we performed single-unit regression using relative threat regressors with assignments of 0.25 and 0.50. The resulting beta coefficients were subjected to ANOVA [factors: assignment (0.25 and 0.50), and interval (1 s bins from 2 s prior to cue onset → 2 s following cue offset)]. Low firing neurons initially

decreased firing according to the actual shock probability (assignment = 0.25), but later decreased firing according to the greater-than-actual, midpoint probability (assignment = 0.50) (Fig. 4b). In support, ANOVA found an assignment × interval interaction ($F_{13,949} = 2.42$, $p = 0.003$, $\eta_p^2 = 0.03$, op = 0.98). Confirming initial signaling of the actual shock probability, early beta coefficients (first 4 s of cue) were shifted below zero for the 0.25 assignment ($M = -0.57$, 95% CI $[-1.02, -0.10]$), but not for the 0.50 assignment ($M = 0.06$, 95% CI $[-0.38, 0.48]$; Fig. 4c, left). Confirming late signaling of the midpoint probability, late beta coefficients (last 2 s cue plus 2 s delay) were shifted below zero for the 0.50 assignment ($M = -0.51$, 95% CI $[-1.03, -0.03]$), but not for the 0.25 assignment ($M = 0.20$, 95% CI $[-0.22, 0.62]$; Fig. 4c, right). Consistent with the ANOVA interaction, there was a positive early-to-late shift in beta coefficients for the 0.25 assignment ($M = 0.76$, 95% CI $[0.41, 1.09]$), but a negative shift for the 0.50 assignment ($M = -0.56$, 95% CI $[-0.95, -0.16]$), and these shifts differed from one another ($M = 1.32$, 95% CI $[0.57, 2.04]$; Fig. 4d).

Intermediate firing neurons also more strongly signaled relative threat compared to fear output (Fig. 4e). Relative threat signaling of the actual probability was superior to an assignment equating uncertainty to safety (assignment = 0.00), but was similar to that of the greater-than-actual, midpoint probability (assignment =

0.50). In support, ANOVA for 0.00 and 0.25 beta coefficients found an assignment × regressor × interval interaction ($F_{13,416} = 2.10$, $p = 0.013$, $\eta_p^2 = 0.06$, op = 0.95). However, ANOVA for 0.25 and 0.50 beta coefficients revealed no assignment × regressor × interval interaction ($F_{13,416} = 1.09$, $p = 0.37$, $\eta_p^2 = 0.03$, op = 0.66).

Early signaling of the actual probability (assignment = 0.25) that switched to signaling of the midpoint probability (assignment = 0.50) was only partially observed in Intermediate firing neurons. ANOVA revealed no assignment × interval interaction ($F_{13,429} = 1.24$, $p = 0.25$, $\eta_p^2 = 0.04$, op = 0.73). Early beta coefficients were shifted below zero for the 0.25 assignment ($M = -0.77$, 95% CI [$-1.50$, $-0.28$]), but not for assignment of 0.50 ($M = 0.07$, 95% CI [$-0.58$, 1.14]; Fig. 4g, left). Late beta coefficients were not shifted from zero for either assignment (0.25: $M = 0.13$, 95% CI [$-0.53$, 0.76]; 0.50: $M = -0.43$, 95% CI [$-1.05$, 0.17]; Fig. 4g, right). There was a positive, early-to-late shift in beta coefficients for the 0.25 assignment ($M = 0.88$, 95% CI [0.32, 1.37]), no shift for assignment of 0.50 ($M = -0.52$, 95% CI [$-1.12$, 0.07]), yet these shifts differed from one another ($M = 1.38$, 95% CI [0.23, 2.39]; Fig. 4h). Of note, ANOVA for Low and Intermediate firing neurons combined [factors: neuron type (Low and Intermediate), assignment (0.25 and 0.50), and interval (1 s bins from 2 s prior to cue onset → 2 s following cue offset)] found an assignment × interval interaction ($F_{13,1378} = 2.52$, $p = 0.002$, $\eta_p^2 = 0.02$, op = 0.98), but no neuron type × assignment × interval interaction ($F_{13,1378} = 0.72$, $p = 0.74$, $\eta_p^2 = 0.01$, op = 0.45). ANOVA did not reveal a significant shift from the actual to the midpoint probability signaling in only Intermediate firing neurons, but their shift did not differ from Low firing neurons.

**Low firing neurons show opposing responses to threat and reward.** While our behavioral procedure is optimized to examine threat-related firing, ongoing reward seeking permitted us to record neural activity around pellet delivery. Although not explicitly cued through the speaker, each reward delivery was preceded by a brief sound caused by the advance of the pellet dispenser. Reward-related firing was extracted from inter-trial intervals, when no cues were presented. We asked if reward-related firing (time locked to pellet dispenser advance) was observed in Low and Intermediate firing neurons. Increases in reward firing—opposing the direction to danger—would indicate relative value signaling that spans reward and threat. The absence of reward firing would indicate specific threat signaling.

To determine reward-related firing, and possible differences between Low and Intermediate firing neurons, we performed repeated measures ANOVA for normalized firing rate [factors: cluster (Low vs. Intermediate) and interval (16 total: 250 ms bins from 2 s prior → 2 s following advancement of pellet dispenser)]. Low firing neurons sharply increased responding following reward delivery, and this firing increase was absent in Intermediate firing neurons (Fig. 5a). In support, ANOVA found a cluster × interval interaction ($F_{15,1560} = 4.75$, $p = 4.57 \times 10^{-9}$, $\eta_p^2 = 0.04$, op = 1.00). Performing separate ANOVA for each cluster revealed a main effect of interval in Low ($F_{15,1065} = 8.07$, $p = 1.51 \times 10^{-17}$, $\eta_p^2 = 0.10$, op = 1.00), but not Intermediate ($F_{15,495} = 1.21$, $p = 0.26$, $\eta_p^2 = 0.04$, op = 0.77) firing neurons. Pre-reward responding by Low firing neurons hovered around zero ($M = -0.06$, 95% CI [$-0.15$, 0.01]), while post-reward firing exceeded pre-reward firing ($M = 0.34$, 95% CI [0.15, 0.53]) and differed from zero ($M = 0.28$, 95% CI [0.14, 0.42]; Fig. 5b). By contrast, pre-reward and post-reward firing never differed from zero for Intermediate firing neurons (pre-reward: $M = 2.60 \times 10^{-4}$, 95% CI [$-0.17$, 0.15]; post-reward: $M = -0.14$, 95% CI [$-0.41$, 0.12]; Fig. 5c).

Not only did Low firing neurons show opposing firing changes to danger and reward, but the firing change was negatively correlated across single units. Low firing neurons showing greater reward firing increases, showed greater danger firing decreases ($R^2 = 0.08$, $p = 0.01$; Fig. 5d). Reward and uncertainty firing were also negatively correlated ($R^2 = 0.16$, $p = 4.63 \times 10^{-4}$; Fig. 5e), but zero firing relationship was observed for reward and safety ($R^2 = 0.006$, $p = 0.51$; Fig. 5f). Even more, equivalent danger-reward and uncertainty-reward correlations were observed in Low firing neurons (Fisher r-to-z transformation, $Z = 0.74$, $p = 0.46$), but uncertainty-reward and safety-reward correlations significantly differed (Fisher r-to-z transformation, $Z = 2.96$, $p = 0.0031$). No cue-reward firing relationships were observed for Intermediate firing neurons (Fig. 5g-i), and these correlations did not differ from one another (all $Z < 1$, all $p > 0.3$). Altogether, the results reveal dynamic signaling of relative threat through firing decreases by VP neurons. Low firing neurons signal general value through opposing responses to threat and reward. Intermediate firing neurons specifically signal relative threat.

**Differential increases in firing are maximal to danger.** We identified 131 neurons (~51% of all cue-responsive neurons) showing firing increases to danger. Cue-excited neurons sharply increased activity at onset, with greatest firing to danger, lesser to uncertainty and least to safety. Differential firing continued during the remainder of the cue and through the 2 s delay period (Fig. 6a). ANOVA for normalized firing rate [factors: cue (danger, uncertainty, and safety) and interval (250 ms bins 2 s prior to cue onset → 2 s following cue offset)] revealed main effects of cue ($F_{2,258} = 68.22$, $p = 1.65 \times 10^{-24}$, $\eta_p^2 = 0.35$, op = 1.00), interval ($F_{55,7095} = 15.06$, $p = 5.42 \times 10^{-130}$, $\eta_p^2 = 0.11$, op = 1.00), and a cue × interval interaction ($F_{110,14190} = 4.52$, $p = 2.31 \times 10^{-49}$, $\eta_p^2 = 0.03$, op = 1.00).

Population-level firing patterns were observed in single units. Firing increases were observed to all cues at onset, but only to the threat cues, danger and uncertainty, in the remaining periods (Fig. 6b). Furthermore, differential firing was observed to every cue pair in every period: danger vs. uncertainty (onset: $M = 0.26$, 95% CI [0.19, 0.34], late cue: $M = 0.24$, 95% CI [0.18, 0.30], and delay: $M = 0.19$, 95% CI [0.13, 0.26]), and uncertainty vs. safety (onset: $M = 0.21$, 95% CI [0.12, 0.29], late cue: $M = 0.15$, 95% CI [0.08, 0.22], and delay: $M = 0.22$, 95% CI [0.14, 0.31]; Fig. 6b). Danger and uncertainty firing were positively correlated for all periods (Supplementary Fig. 8a-c). By contrast, positively correlated firing to uncertainty and safety at cue onset gave way to zero correlation during late cue and negatively correlated during the delay period (Supplementary Fig. 8d-f). Cessation of nose poking in the absence of cues was insufficient to increase firing (Supplementary Fig. 9).

**Cue-excited neurons signal relative threat and fear output.** Of course, descriptive firing analyses cannot distinguish between relative threat and fear output signaling. To do this we performed single-unit, linear regression (described above). Across all assignments, single-unit activity was captured by a mixture of relative threat and fear output (Supplementary Fig. 10). Relative threat signaling was better captured by the midpoint shock probability (assignment = 0.50), compared to the actual probability (assignment = 0.25) or a higher probability (assignment = 0.75). In support, ANOVA for beta coefficients [factors: assignment (0.25 and 0.50), regressor (fear output and relative threat), and interval (1 s bins from cue onset → 2 s following cue offset)] revealed an assignment × regressor interaction ($F_{1,129} = 6.20$, $p = 0.014$, $\eta_p^2 = 0.05$, op = 0.70). Similarly, ANOVA for 0.50 and

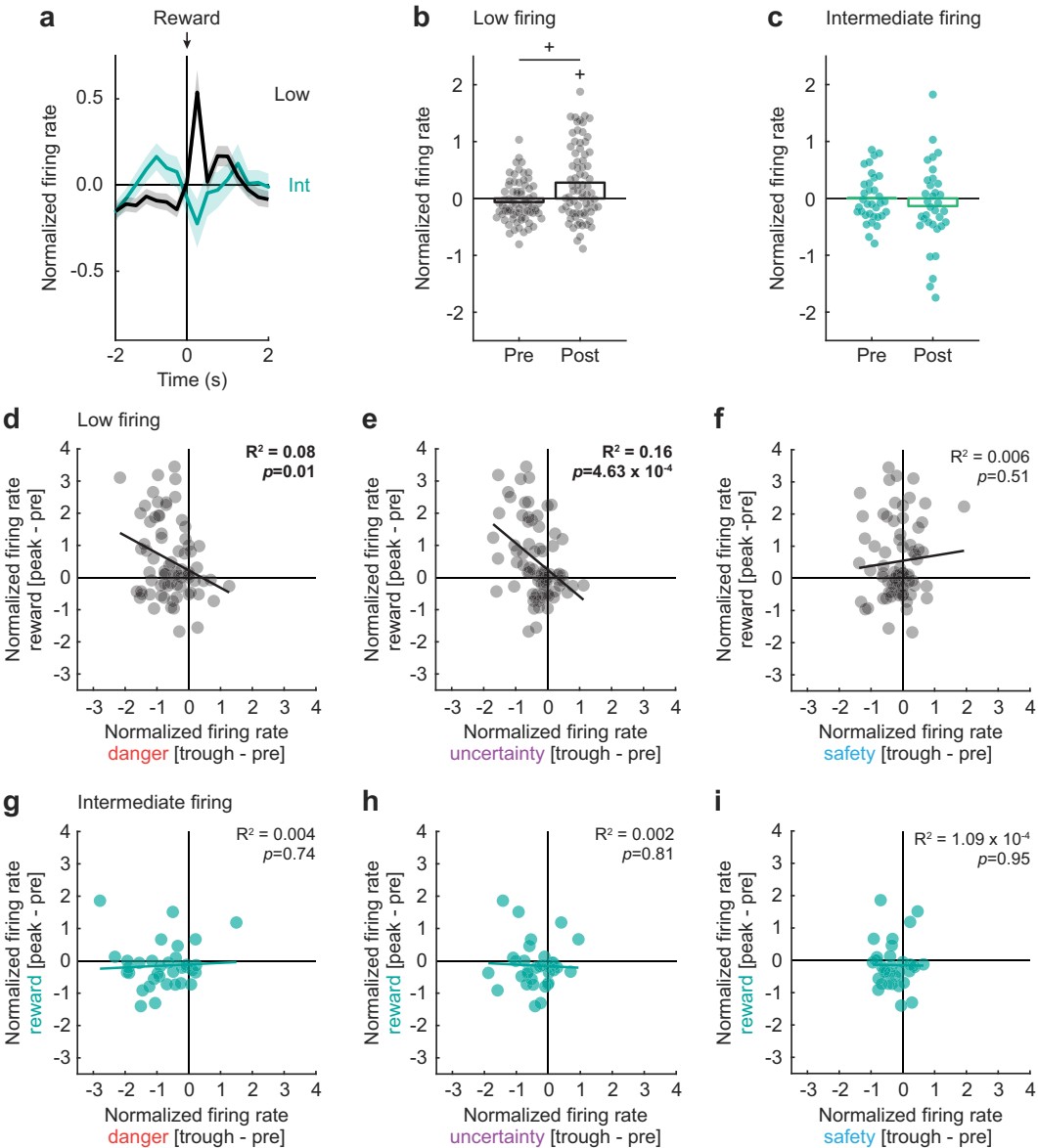

**Fig. 5 Low firing neurons show opposing responses to threat and reward. a** Mean ± SEM normalized firing rate to reward is shown 2 s prior to and 2 s after reward delivery (advancement of pellet dispenser) for the Low ($n = 74$, black) and Intermediate ($n = 34$, turquoise) firing neurons. Reward delivery is indicated by black arrow. SEM is indicated by shading. **b** Mean (bar) and individual (data points), normalized firing rate for Low firing neurons are shown during 500 ms interval prior (pre) to and 500 ms interval after (post) reward delivery. **c** Identical graph made for the Intermediate firing neurons, as in **b**. **d** Mean normalized firing rate to reward (250 ms prior to reward delivery to 250 ms following reward delivery, [peak - pre]) vs. danger (the second 250 ms of cue, [trough - pre], red) is plotted for Low firing neurons (black). **e, f** Mean normalized firing rate to **e** reward vs. uncertainty (purple) and **f** reward vs. safety (blue) is plotted for Low firing neurons, as in **d**. Trendline, the square of the Pearson correlation coefficient ($R^2$) and associated $p$ value ($p$) are shown for each graph. **g–i** Mean normalized firing rate to **g** reward vs. danger, **h** reward vs. uncertainty, and **i** reward vs. safety for Intermediate firing neurons (turquoise), as in **d–f**. [+]95% bootstrap confidence interval for differential reward firing does not contain zero. [+]95% bootstrap confidence interval for normalized firing rate does not contain zero.

0.75 beta coefficients revealed an assignment × regressor interaction ($F_{1,129} = 3.92$, $p = 0.05$, $\eta_p^2 = 0.03$, op = 0.50).

Early signaling of elevated relative threat gave way to joint signaling of relative threat and fear output until shock delivery (Fig. 6c). ANOVA for beta coefficients [factors: regressor (0.50 assignment and fear output) and interval (1 s bins from 2 s prior to cue onset → 2 s following cue offset)] found a main effect of regressor ($F_{1,128} = 6.29$, $p = 0.013$, $\eta_p^2 = 0.05$, op = 0.70) and a regressor × interval interaction ($F_{13,1664} = 2.15$, $p = 0.01$, $\eta_p^2 = 0.02$, op = 0.96). Beta coefficients exceeding zero were observed for relative threat (assignment = 0.50) at cue onset ($M = 0.59$, 95% CI [0.41, 0.76]), as well as relative threat and fear output

during late cue (0.50: $M = 0.38$, 95% CI [0.21, 0.54]; FO: $M = 0.19$, 95% CI [0.08, 0.30]) and delay (0.50: $M = 0.43$, 95% CI [0.26, 0.61]; FO: $M = 0.17$, 95% CI [0.05, 0.29]; Fig. 6d). Relative threat and fear output beta coefficients differ from one another during onset ($M = 0.48$, 95% CI [0.20, 0.75], but not during the late cue ($M = 0.19$, 95% CI [−0.03, 0.44]), and delay ($M = 0.27$, 95% CI [−0.007, 0.56]; Fig. 6d).

**Cue-excited neurons increase firing to reward**. We examined reward-related firing, time locked to the pellet dispenser advance, to determine if cue-excited neurons specifically signaled threat or

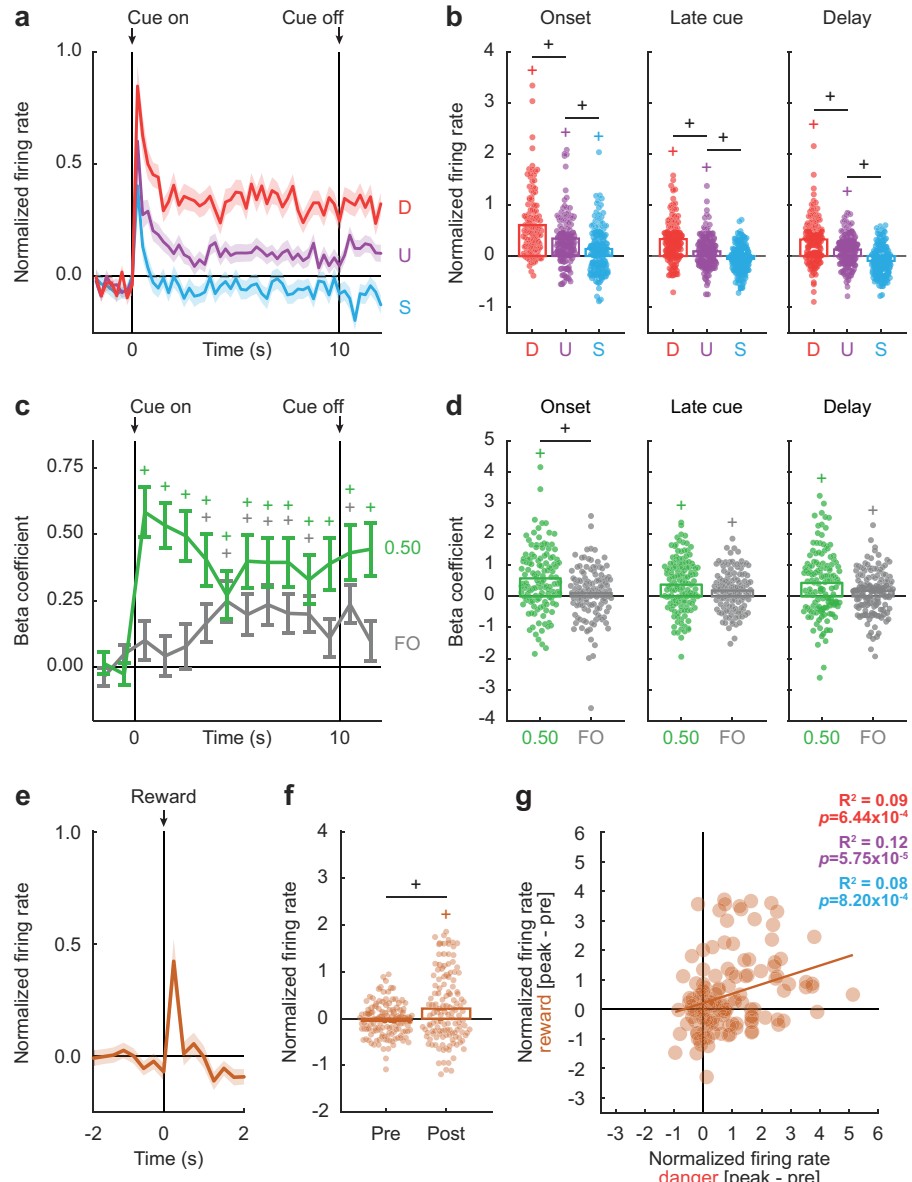

**Fig. 6 Differential cue firing by cue-excited neurons reflects relative threat and fear output. a** Mean normalized firing rate to danger (D, red), uncertainty (U, purple), and safety (S, blue) is shown from 2 s prior to cue onset to 2 s following cue offset for the cue-excited population ($n = 131$). Cue onset and offset are indicated by vertical black lines. **b** Mean (bar) and individual (data points), normalized firing rate for cue-excited neurons during the first 1 s cue interval (onset, left), the last 5 s cue interval (late cue, middle), and 2 s following cue offset (delay, right) are shown for each cue (D, danger, red; U, uncertainty, purple; S, safety, blue). $^+$95% bootstrap confidence interval for differential cue firing rate does not contain zero. $^+$95% bootstrap confidence interval for normalized firing rate does not contain zero (colored plus signs). **c** Mean ± SEM beta coefficients are shown for each regressor (0.50, assignment, green; FO, fear output, gray), from 2 s prior to cue onset to 2 s following cue offset in 1 s intervals, for the cue-excited population. Cue onset and offset are indicated by vertical black lines. **d** Mean (bar) and individual (data points), beta coefficient for each regressor (0.50, assignment, green; FO, fear output, gray) for cue-excited neurons. $^+$95% bootstrap confidence interval for differential beta coefficient does not contain zero. $^+$95% bootstrap confidence interval for beta coefficient does not contain zero (colored plus signs). **e** Mean normalized firing rate to reward is shown 2 s prior to and 2 s after the reward delivery (advancement of pellet dispenser) for the cue-excited population (maroon). Reward delivery is indicated by vertical black line. **f** Mean (bar) and individual (data points), normalized firing rate for cue-excited neurons are shown during 500 ms interval prior (pre) to and 500 ms interval after (post) the reward delivery. $^+$95% bootstrap confidence interval for differential firing rate does not contain zero. $^+$95% bootstrap confidence interval for firing rate does not contain zero (maroon plus sign). **g** Mean normalized firing rate to reward (250 ms prior to reward delivery to 250 ms following reward delivery, [peak - pre]) vs. danger (the first 250 ms of cue, [peak - pre], red) is plotted for cue-excited neurons. Trendline, the square of the Pearson correlation coefficient ($R^2$) and associated $p$ value ($p$) are shown. $R^2$ and $p$ for uncertainty (purple) and safety (blue) are provided.

signaled salience. In support of salience signaling, cue-excited neurons increased firing to reward (Fig. 6e). Repeated measures ANOVA for normalized firing rate [factor: interval (16 total: 250 ms bins from 2 s prior → 2 s following advancement of pellet dispenser)] revealed a main effect of interval ($F_{15,1920} = 7.24$, $p =$

$9.32 \times 10^{-16}$, $\eta_\mathrm{p}^2 = 0.05$, op = 1.00). Single-unit firing prior to reward delivery hovered around zero. Firing increases following reward delivery exceeded zero ($M = 0.22$, 95% CI [0.07, 0.35]), and also exceeded pre-reward firing ($M = 0.26$, 95% CI [0.10, 0.41]; Fig. 6f). Further supporting salience signaling, the

magnitude of firing increase to each cue and reward was positively correlated. Single units showing greater firing increases at reward peak, showed greater firing increases at danger ($R^2 = 0.09$, $p = 6.44 \times 10^{-4}$), uncertainty ($R^2 = 0.12$, $p = 5.75 \times 10^{-5}$), and safety peak ($R^2 = 0.08$, $p = 8.20 \times 10^{-4}$; Fig. 6g). Finally, peak danger firing ($M = 0.40$, 95% CI [0.13, 0.65]), but not peak uncertainty ($M = 0.13$, 95% CI [$-0.07$, 0.35]), and safety ($M = -0.08$, 95% CI [$-0.30$, 0.13]) firing, differed from peak reward firing. Though showing differential firing increases to danger, uncertainty, and safety; comparable firing increases to reward support salience signaling by cue-excited neurons.

## Discussion

We recorded VP single-unit activity while rats discriminated danger, uncertainty and safety. Revealing widespread threat-related firing, most VP neurons were maximally responsive to danger. Cue-inhibited neuron types (Low firing and Intermediate firing neurons) dynamically signaled relative threat. Initial relative threat signaling was precise, with neurons decreasing cue firing in proportion to the actual foot shock probability. As foot shock drew near, firing decreases disproportionate to the uncertainty foot shock probability were observed. Low firing neurons increased activity following reward delivery, marking these neurons as a possible source of relative value that spans threat and reward. Intermediate firing neurons more exclusively signaled relative threat. Consistent with salience signaling, another VP population showing cue firing increases, also increased firing to reward. Cue firing by salience neurons reflected fear output, and relative threat that was disproportionate to foot shock probability.

Before discussing our results, some considerations must be raised. The first concerns biological sex. We limited this study to males, in part to enable comparison of VP responding to prior reports, which have been mostly in males[28,31,34–36,38,40,41,46,55–57]. So far, no differences in VP activity/function have been found in studies that examined biological sex[19,42,43]. Our laboratory has observed complete and comparable fear discrimination in male and female rats[11,58–61]. We predict that equivalent relative threat signaling will be observed in female and male VP neurons. Our observation of dynamic relative threat signaling in male rats permits a direct test of this hypothesis in future studies. Another consideration is that our behavioral design did not manipulate reward with the same nuance as threat. This was intentional, as our goal was to examine threat behavior, firing and signaling. Nevertheless, our design prevents a definitive demonstration of relative value signaling that spans threat and reward. Such a demonstration would require a behavioral procedure in which 5+ cues predict unique shock and reward probabilities, observing complete behavioral discrimination.

Low and Intermediate firing neurons dynamically signaled relative threat, rather than fear output (via conditioned suppression). We interpret the lack of fear output signaling to be meaningful because our regression analysis was able to detect fear output signaling in cue-excited VP neurons. The same regression approach has also shown fear output signaling through firing increases and decreases in the ventrolateral periaqueductal gray[58,62]. Of course, VP neurons could signal fear output through a different behavioral mechanism, such as freezing. VP signaling of freezing may be unlikely given that conditioned freezing and conditioned suppression are correlated in intact rats[48]. Even if not freezing, many more signals for fear output are possible: bradycardia[63], micturition[63], piloerection[64], change in body temperature[65], hyperventilation[66], etc. Our results cannot absolutely reject VP fear output signaling, but do stipulate that any output signaled must closely map to foot shock probability, confounding fear output with relative threat signaling.

The patterned activity of Low firing neurons is broadly consistent with studies showing VP populations with opposing changes in firing to reward and aversive cues. In one study, mice were trained to associate unique auditory cues with outcomes of differing valence (water vs. air puff) and size (small vs. large)[19]. Mice showed differential licking to the large and small water cues (large > small), and VP GABA neurons showed differential firing increases to water cues based on their size (large > small). These same neurons showed firing decreases to air puff cues that less clearly differentiated size (large~small). However, behavior around air puff was not measured, so the lack of differential firing decreases may have resulted from a lack of behavioral discrimination. In the most recent study, monkeys were trained to associate visual cues with liquid reward, air puff or nothing (neutral)[44]. One VP population showed firing increases to the liquid reward cue, but firing decreases to the air puff cue. Yet, these same neurons showed comparable firing decreases to the neutral cue and air puff cue. Behaviorally, monkeys treated the neutral cue more similarly to the air puff cue. Our results suggest that differential VP firing decreases may be most apparent when threat behavior is explicitly examined, or when complete behavioral discrimination of neutral and aversive cues is observed. Building on these studies, our findings reveal that Low firing VP neurons, putative GABA output neurons[19,43,67], signal relative value that spans reward and threat.

Relative threat signaling through VP firing decreases is readily integrated into neural circuits permitting fine tuning of threat behavior. Previous work has shown that suppressing VP activity promotes aversive behavior. So while optogenetic activation of all VP neurons/VP GABA neurons induces place preference[19,21,43], inhibition of VP GABA neurons induces place aversion[43]. VP GABA firing decreases may simultaneously modulate ventral tegmental area (VTA)-driven reward behavior and BLA-driven threat behavior. VP GABA neurons directly project to dopamine neurons and GABA interneurons in the VTA[21,43,68]. GABA neurons comprise ~25% of the VP input to the BLA[22,43,69]. Consistent with a previous proposal[70], threat-induced VP GABA firing decreases may increase VTA GABA activity, suppressing VTA dopamine firing to reduce reward behavior. At the same time, threat-induced VP GABA firing decreases may disinhibit BLA firing to promote threat behavior. By scaling firing decreases to degree of threat, VP neurons may precisely modulate VTA and BLA firing, thereby controlling the level of the threat response.

Low/Intermediate firing neurons may also include cholinergic neurons, which comprise ~75% of the VP input to the BLA[71,72], or may even include proenkephalin neurons[73]. Stimulating basal forebrain cholinergic terminals in the BLA inhibits principal neurons that are modestly depolarized or at rest[74], permitting VP firing pauses may act to promote BLA firing. Exciting VP proenkephalin neurons reduces inhibitory avoidance[73], suggesting VP firing pauses may promote inhibitory avoidance. Thus, VP cholinergic and GABAergic firing decreases—dynamically signaling relative threat—may be positioned to suppress VTA dopamine firing and promote BLA firing to precisely tune threat behavior.

We recorded VP single-unit activity during fear discrimination with foot shock outcome to explicitly examine threat signaling. We found that VP neurons did not merely reduce firing to threat cues, but scaled firing decreases in proportion to the foot shock probability associated with each cue. Firing decreases were better captured by relative threat, compared to the rat's fear behavior. Precise, relative threat signaling was most evident during early cue presentation. Interestingly, firing decreases proportional to foot shock probability gave way to firing decreases that were disproportionately large to uncertain threat as foot shock drew near. Our results suggest the VP is a source of a dynamic threat

signal that aligns defensive behavior to threat probability when threat is distal, but promotes disproportionate defensive behavior when threat is imminent. Detailing how VP threat signals shape activity in the BLA and VTA, and across a brain-wide threat network, is likely to provide insight into the neural basis of adaptive and maladaptive threat behavior.

## Methods

The VP recording/fear discrimination approach is based on prior work from our laboratory[58,59,62].

**Experimental subjects**. A total of 14 adult male Long Evans rats, weighing 250–275 g were obtained from Long Evans breeders maintained in the Boston College Animal Care Facility. The rats were single-housed on a 12 h light/dark cycle (lights on at 7:00 a.m.) with free access to water. Rats were maintained at 85% of their free-feeding body weight with standard laboratory chow (18% Protein Rodent Diet #2018, Harlan Teklad Global Diets, Madison, WI), except during surgery and post-surgery recovery. All protocols were approved by the Boston College Animal Care and Use Committee and all experiments were carried out in accordance with the NIH guidelines regarding the care and use of rats for experimental procedures.

**Electrode assembly**. Microelectrodes consisted of a drivable bundle of sixteen 25.4 μm diameter Formvar-Insulated Nichrome wires (761500, A-M Systems, Carlsborg, WA) within a 27-gauge cannula (B000FN3M7K, Amazon Supply) and two 127 μm diameter PFA-coated, annealed strength stainless-steel ground wires (791400, A-M Systems, Carlsborg, WA). All wires were electrically connected to a nano-strip Omnetics connector (A79042-001, Omnetics Connector Corp., Minneapolis, MN) on a custom 24-contact, individually routed and gold immersed circuit board (San Francisco Circuits, San Mateo, CA). Sixteen individual recording wires were soldered to individual channels of an Omnetics connector. The sixteen wire bundle was integrated into a microdrive permitting advancement in ~42 μm increments.

**Surgery**. Stereotaxic surgery was performed aseptic conditions under isoflurane anesthesia (1–5% in oxygen). Carprofen (5 mg/kg, i.p.) and lactated ringer's solution (10 ml, s.c.) were administered preoperatively. The skull was scoured in a crosshatch pattern with a scalpel blade to increase efficacy of implant adhesion. Six screws were installed in the skull to further stabilize the connection between the skull, electrode assembly and a protective head cap. A 1.4 mm diameter craniotomy was performed to remove a circular skull section centered on the implant site and the underlying dura was removed to expose the cortex. Nichrome recording wires were freshly cut with surgical scissors to extend ~2.0 mm beyond the cannula. Just before implant, current was delivered to each recording wire in a saline bath, stripping each tip of its formvar insulation. Current was supplied by a 12 V lantern battery and each Omnetics connector contact was stimulated for 2 s using a lead. Machine grease was placed by the cannula and on the microdrive. For implantation dorsal to the VP, the electrode assembly was slowly advanced (~100 μm/min) to the following coordinates: −0.08 mm form bregma, −2.05 mm lateral from midline, and −6.95 mm ventral from the cortex. Once in place, stripped ends of both ground wires were wrapped around two screws in order to ground the electrode. The microdrive base and a protective head cap were cemented on top of the skull using orthodontic resin (C 22-05-98, Pearson Dental Supply, Sylmar, CA), and the Omnetics connector was affixed to the head cap.

**Behavioral apparatus**. All experiments were conducted in two, identical sound-attenuated enclosures that each housed a Pavlovian fear discrimination chamber with aluminum front and back walls retrofitted with clear plastic covers, clear acrylic sides and top, and a stainless steel grid floor. Each grid floor bar was electrically connected to an aversive shock generator (Med Associates, St. Albans, VT) through a grounding device. This permitted the floor to be grounded at all times except during shock delivery. An external food cup and a central nose poke opening, equipped with infrared photocells were present on one wall. Auditory stimuli were presented through two speakers mounted on the ceiling of enclosure. Behavior chambers were modified to allow for free movement of the electrophysiology cable during behavior; plastic funnels were epoxied to the top of the behavior chambers with the larger end facing down, and the tops of the chambers were cut to the opening of the funnel.

**Nose poke acquisition**. Experimental procedure started with two days of pre-exposure in the home cage where rats received the pellets (Bio-Serv, Flemington, NJ) used for rewarded nose poking. Rats were then shaped to nose poke for pellet delivery in the behavior chamber using a fixed ratio schedule in which one nose poke yielded one pellet until they reached at least 50 nose pokes. Over the next 5 days, rats were placed on variable interval (VI) schedules in which nose pokes were reinforced on average every 30 s (VI-30, day 1), or 60 s (VI-60, days 2 through

5). For fear discrimination sessions, nose pokes were reinforced on a VI-60 schedule independent of auditory cue or foot shock presentation.

**Fear discrimination**. Prior to surgery, each rat received eight 54-min Pavlovian fear discrimination sessions. Each session consisted of 16 trials, with a mean inter-trial interval of 3.5 min. Auditory cues were 10 s in duration and consisted of repeating motifs of a broadband click, phaser, or trumpet (listen or download: http://mcdannaldlab.org/resources/ardbark). Each cue was associated with a unique probability of foot shock (0.5 mA, 0.5 s): danger, $p = 1.00$; uncertainty, $p = 0.25$; and safety, $p = 0.00$. Auditory identity was counterbalanced across rats. For danger and uncertainty shock trials, foot shock was administered 2 s following the termination of the auditory cue. This was done in order to observe possible neural activity during the delay period is not driven by an explicit cue. A single session consisted of four danger trials, two uncertainty shock trials, six uncertainty omission trials, and four safety trials. The order of trial type presentation was randomly determined by the behavioral program, and differed for each rat, each session. After the eighth discrimination session, rats were given full food and implanted with drivable microelectrode bundles. Following surgical recovery, discrimination resumed with single-unit recording. The microelectrode bundles were advanced in ~42–84 μm steps every other day to record from new units during the following session.

**Single-unit data acquisition**. During recording sessions, a 1× amplifying head-stage connected the Omnetics connector to the commutator via a shielded recording cable (Headstage: 40684-020 and Cable: 91809-017, Plexon Inc., Dallas TX). Analog neural activity was digitized and high-pass filtered via amplifier to remove low-frequency artifacts and sent to the Ominplex D acquisition system (Plexon Inc., Dallas TX). Behavioral events (cues, shocks, nose pokes, and pellet deliveries) were controlled and recorded by a computer running Med Associates software. Timestamped events from Med Associates were sent to Ominplex D acquisition system via a dedicated interface module (DIG-716B). The result was a single file (.pl2) containing all time stamps for recording and behavior. Single units were sorted offline with a template-based spike-sorting algorithm (Offline Sorter V3, Plexon Inc., Dallas TX). Timestamped spikes and events (cues, shocks, nose pokes, and pellet deliveries) were extracted and analyzed with statistical routines in Matlab (Natick, MA).

**Histology**. Rats were deeply anesthetized using isoflurane and current from a 6 V battery was passed through 4 of the 16 nichrome electrode wires. Rats were transcardially perfused with 0.9% biological saline and 4% paraformaldehyde in a 0.2 M Potassium Phosphate Buffered solution. Brains were extracted and post-fixed in a 10% neutral-buffered formalin solution for 24 h, stored in 10% sucrose/formalin, frozen at −80 °C and sectioned via sliding microtome. In order to identify VP boundaries, we performed immunohistochemistry for substance P (primary antibody, rabbit anti-substance P, 1:100, Immunostar, Hudson, WI; secondary antibody, Alexa Fluor 594 donkey anti-rabbit, Jackson ImmunoResearch Laboratories, West Grove, PA), and NeuroTrace™ (1:200, Thermo Fisher Scientific, Waltham, MA). Sections were mounted on coated glass slides, coverslipped with Vectashield mounting medium without DAPI (Vector Laboratories, Burlingame, CA), and imaged using a fluorescent microscope (Axio Imager Z2, Zeiss, Thornwood, NY).

**Verifying electrode placement**. Passing current through the wire permitted the tip locations to be observed in brain sections. In addition, wire tracks leading up to tips were visible. Starting with the electrode tips, the driving path of the electrode through the brain was backwards calculated. Only recording locations below the anterior commissure and inside of the dense substance P field were considered to be in the VP (Fig. 1d). Only single units recorded from sites within the boundaries of VP (Fig. 1e) were included in analyses[75].

**95% bootstrap confidence interval**. 95% bootstrap confidence intervals were constructed for suppression ratios, normalized firing rate, and beta coefficients using the bootci function in Matlab. For each bootstrap, a distribution was created by sampling the data 1000 times with replacement. Studentized confidence intervals were constructed with the final outputs being the mean, lower bound and upper bound of the 95% bootstrap confidence interval. Differential suppression ratios, firing rates, and beta coefficients were said to be observed when the 95% confidence interval did not include zero.

**Calculating suppression ratios**. Fear was measured by suppression of rewarded nose poking, calculated as a ratio: [(baseline poke rate − cue poke rate)/(baseline poke rate + cue poke rate)]. The baseline nose poke rate was taken from the 20 s prior to cue onset and the cue poke rate from the 10 s cue period. Suppression ratios were calculated for each trial using only that trial's baseline. A ratio of '1' indicated high fear, '0' low fear, and gradations between intermediate levels of fear. Suppression of rewarded nose poking was used because it is an objective, continuous measure of fear output[50]. Suppression ratios permit observation of differential fear to danger, uncertainty and safety; as well as precise, single trial fear

measures necessary for single-unit regression. Suppression ratios were analyzed using ANOVA with cue (danger, uncertainty, and safety) as a factor (Fig. 1f). Uncertainty shock and uncertainty omission trials were collapsed because they did not differ for suppression ratio; during cue presentation, rats did not know the current uncertainty trial type. F statistic, p value (p), partial eta squared ($\eta_p^2$) and observed power (op) are reported for significant main effects and interactions. The distribution of suppression ratios was visualized using the plotSpread function for Matlab (https://www.mathworks.com/matlabcentral/fileexchange/37105-plot-spread-points-beeswarm-plot).

**Identifying cue-responsive neurons**. Single units were screened for cue responsiveness by comparing mean firing rate (Hz) during the 10 s prior to cue presentation (baseline), to mean firing rate (Hz) during the first 1 s and last 5 s of cue presentation. A neuron was considered cue-responsive if it showed a significant change in firing from baseline (increase or decrease; paired, two-tailed t-test, $p < 0.05$) to danger, uncertainty or safety during the first 1 s or the last 5 s interval. A Bonferroni correction (0.5/6) was not performed because this criterion was too stringent, resulting in many cue-responsive neurons being omitted from analysis.

**Firing and waveform characteristics**. The following characteristics were determined for each cue-responsive neuron: baseline firing rate, coefficient of variance, coefficient of skewness, waveform half duration, and waveform amplitude ratio (Supplementary Fig. 3). Baseline firing rate was mean firing rate (Hz) during the 10 s prior to cue onset. Coefficient of variance was calculated by [$SD_{ISI}/\overline{X}_{ISI}$], in which $SD_{ISI}$ was the standard deviation of inter-spike interval, and $\overline{X}_{ISI}$ was the mean inter-spike interval. Coefficient of variance is a relative measure of the variability of spike firing, with small values indicating less variation in inter-spike intervals (more regular firing), and large values more variability (less regular firing)[52,53]. Coefficient of skewness was calculated by [$(3 \times (\overline{X}_{ISI} - \tilde{X}_{ISI}))/SD_{ISI}$], in which $\overline{X}_{ISI}$, $\tilde{X}_{ISI}$, and $SD_{ISI}$ were the mean, median and standard deviation of inter-spike interval, respectively. Coefficient of skewness is a measure of the asymmetry of the distribution of the inter-spike intervals, with positive values indicating longer intervals (less regular firing) and negative values indicating shorter intervals (more regular firing)[53]. Waveform amplitude ratio was calculated by [$(N - P)/(N + P)$], in which P was the y-axis distance between the initial value and peak initial hyperpolarization, and N was the y-axis distance between the peak initial value and valley of depolarization. Values near zero indicate a relatively large initial hyperpolarization while values near one indicate a relatively small initial hyperpolarization[54,62]. Waveform half duration was calculated by [$D/2$], in which D was the x-axis distance between the valley of depolarization and the peak of after-hyperpolarization and smaller values indicate narrower waveforms[54,62].

**K-means clustering**. We used k-means clustering to identify cue-inhibited sub-populations. Clustering was performed using the Matlab kmeans function. K-means clustering used baseline firing rate and four additional characteristics (coefficient of variance, coefficient of skewness, waveform half duration, and waveform amplitude ratio) and identified three clusters within the population.

**Z-score normalization**. For each neuron, and for each trial type, firing rate (Hz) was calculated in 250 ms bins from 20 s prior to cue onset to 20 s following cue offset, for a total of 200 bins. Mean firing rate over the 200 bins was calculated by averaging all trials for each trial type. Mean differential firing was calculated for each of the 200 bins by subtracting mean baseline firing rate (10 s prior to cue onset), specific to that trial type, from each bin. Mean differential firing was Z-score normalized across all trial types within a single unit, such that mean firing = 0, and standard deviation in firing = 1. Z-score normalization was applied to firing across the entirety of the recording epoch, as opposed to only the baseline period, in case neurons showed little/no baseline activity. As a result, periods of phasic, excitatory, and inhibitory firing contributed to normalized mean firing rate (0). For this reason, Z-score normalized baseline activity can differ from zero. Z-score normalized firing was analyzed with ANOVA using cue, and bin as factors. F statistic and p values (p) are reported, as well as partial eta squared ($\eta_p^2$) and observed power (op). Reward-related firing was extracted from inter-trial intervals, when no cues were presented. Although not explicitly cued through the speaker, each reward delivery was preceded by a brief sound caused by the advance of the pellet dispenser. For reward-related firing (time locked to pellet dispenser advance), firing rate (Hz) was calculated in 250 ms bins from 2 s prior to 2 s following advancement of pellet dispenser, for a total of 16 bins. Mean differential firing was calculated for each of the 16 bins by subtracting pre-reward firing rate (mean of 1 s prior to reward delivery).

**Heat plot and color maps**. Heat plots were constructed from normalized firing rate using the imagesc function in Matlab (Fig. 2). Perceptually uniform color maps were used to prevent visual distortion of the data[76].

**Population and single-unit firing analyses**. Population cue firing was analyzed using ANOVA with cue (danger, uncertainty, and safety) and interval (250 ms bins from 2 s prior to cue onset to 2 s following cue offset) as factors (Fig. 3,

Fig. 6a, b, and Supplementary Fig. 4a, b). Uncertainty trial-types were collapsed because they did not differ firing analysis. This was expected, during cue presentation rats did not know the current uncertainty trial-type. F statistic, p value (p), partial eta squared ($\eta_p^2$), and observed power (op) are reported for main effects and interactions. The 95% bootstrap confidence intervals were reconstructed for normalized firing to each cue (compared to zero), as well as for differential firing (danger vs. uncertainty) and (uncertainty vs. safety), during cue onset (first 1 s cue interval), late cue (last 5 s cue interval), and delay (2 s following cue offset) periods. The distribution of single-unit firing was visualized using a plotSpread function for Matlab. Population reward firing was analyzed using repeated measures ANOVA with interval (250 ms bins from 2 s prior to 2 s following advancement of pellet dispenser) as factor (Fig. 5a–c, Fig. 6e, f, and Supplementary Fig. 4c, d). The 95% bootstrap confidence intervals were reconstructed for normalized firing rate to reward during pre (500 ms prior to reward delivery), and post (first 500 ms following reward delivery) (compared to zero), as well as for differential firing (pre vs. post). Relationships between cue firing (danger vs. uncertainty, and uncertainty vs. safety; Supplementary Fig. 5 and Supplementary Fig. 8), as well as between reward and cue firing (Fig. 5d–i, Fig. 6g, and Supplementary Fig. 4e) were determined by calculating the $R^2$ and p value (p) for the Pearson's correlation coefficient. Population firing was analyzed using repeated measures ANOVA with interval (250 ms bins from 2 s prior to 2 s following nose poke cessation) as factor (Supplementary Fig. 6 and Supplementary Fig. 9). The 95% bootstrap confidence intervals were reconstructed for normalized firing rate during pre (500 ms prior to nose poke cessation), and post (first 500 ms following nose poke cessation) (compared to zero), as well as for differential firing (pre vs. post).

**Single-unit linear regression**. Single-unit, linear regression was used to determine the degree to which fear output and relative threat explained trial-by-trial variation in firing of single units in a specific time interval. For each regression, all 16 trials from a single session were ordered by type. Z-score normalized firing rate was specified for the interval of interest. The fear output regressor was the suppression ratio for the entire cue, for that specific trial. The relative threat regressor assigned values to each trial type. The values for danger (1) and safety (0) were fixed. The value assigned to uncertainty was systematically increased from 0 to 1 in 0.25 steps (0.00, 0.25, 0.50, 0.75, and 1.00). Regression (using the regress function in Matlab) required a separate, constant input. The regression output was the beta coefficient for each regressor (relative threat and fear output), quantifying the strength (greater distance from zero = stronger) and direction (>0 = positive) of the predictive relationship between each regressor and single-unit firing (Supplementary Fig. 7 and Supplementary Fig. 10). ANOVA was used to analyze beta coefficients, exactly as described for normalized firing rate (Fig. 4 and Fig. 6c, d). 95% bootstrap confidence intervals were reconstructed for beta coefficients (compared to zero), as well as for relative threat vs. fear output. The distribution of single-unit beta coefficients visualized using a plotSpread function for Matlab. In the case of Low and Intermediate firing neurons, single-unit firing was equivalently captured by relative threat using assignments of 0.25 and 0.50. Regression was performed as above, only now using two relative threat regressors with assignments of 0.25 and 0.50. 95% bootstrap confidence intervals performed as described above.

**Additional resources**. Med Associates programs used for behavior and Matlab programs used for behavioral analyses are made freely available at our lab website: http://mcdannaldlab.org/resources

**Reporting summary**. Further information on research design is available in the Nature Research Reporting Summary linked to this article.

## Data availability
Full electrophysiology data set will be uploaded to http://crcns.org/ upon acceptance for publication.

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

## Acknowledgements

Research reported in this publication was supported by the National Institute of Mental Health of the National Institutes of Health under Award Numbers MH113053 and MH117791. The content is solely the responsibility of the authors and does not necessarily represent the official views of the National Institutes of Health. We thank Bret Judson and the Boston College Imaging Core for infrastructure and support.

## Author contributions

M.H.R. bred the rats, M.M. and M.A.M. designed the experiment, M.M. performed the surgeries and M.M. collected the single-unit and behavioral data. M.M. and M.A.M. interpreted/analyzed the data and wrote the manuscript, M.M., M.A.M., and M.H.R. edited and approved the final manuscript.

## Competing interests

The authors declare no competing interests.
