## [Peer Review File · Communications Biology]

Reviewers' comments:

Reviewer #1 (Remarks to the Author):

Ventral pallidum is well known for its role in reward related behavior. Less is known about how this structure participates in threat. The authors used a conditioned suppression task while recording from VP neurons. VP neurons were highly sensitive to danger cues, with about half of neurons increase or decreasing. Baseline firing was not informative for cue increasing neurons but varied with decreasing neurons. Profiles of low, intermediate, and high baseline firing differed with respect to decreasing cue type. Low and intermediate decreased firing neurons signaled relative threat. Low firing decreasing neurons differently fired for danger cues and reward, others did not. Cue increasing neurons were scaled to danger and increased to reward, suggesting salience signaling. The experiments are well done with robust statistical analyses. My major concerns are with data interpretation.

Major concerns:

1. Abstract writes an "integral role for the VP in threat-related behavior" but no test of this role was performed.
2. It was not clear how the authors could tell where their recording electrodes were located, at least in the picture shown in figure 1D. How can the authors determine track from recording location?
3. The analysis of single neurons and trials as subjects using t-tests violates the assumption of independence.
4. It was not clear if reward related analyses were at the time of reward consumption or the click sound.
5. The analysis on fear output signaling would be bolstered if video analyses confirmed freezing behavior related to the change in firing.
6. The discussion linking recorded cell-types to genetically defined neurons (interneurons, Penk, cholinergic, projections to specific targets, etc) is highly speculative without supporting data. Suggest remove much of this from the discussion.

Minor concerns:

1. Authors should be skeptical of literature claiming central amygdala projections to VP given that the extended amygdala system projects into the substantia innominata and other structures close to VP. Classic studies from Zahm, Heimer, and Alheid suggested the extended amygdala was largely distinct from the VP within the mesolimbic system.
2. The sentence in the discussion regarding VP glutamate and GABA projections to amygdala - my knowledge of this subject, which may be limited, is that the cholinergic neurons project to amygdala primarily.

Reviewer #2 (Remarks to the Author):

This work by Moaddab and colleagues presents evidence from electrophysiology that VP neurons are poised to encode relative threat values. The study seems rigorous, the behavioral task elegant and the findings point to a role for VP neurons in signaling cues predicting aversive outcomes. There are a few points regarding statistical analysis (noted below) and graphical representation of results would benefit from clarification. The strength of the study would also be bolstered by contextualization of the findings, and a discussion of the novelty of the authors results compared to prior work that has established the VP as important for signaling aversive outcomes and punishment.

The introduction does not well-contextualize the current state of the literature about the role of VP in reward and aversion, particularly given the known neurochemical heterogeneity that has been mapped on to functional heterogeneity (Knowland et al., Cell 2017, Faget et al., Nat Comm 2018, Tooley et al., Biol Psych 2018). Discussion of VP populations without mention of functional, anatomical or neurochemical specificity make it difficult to follow the logic of the introduction. Are the authors arguing that VP neurons are excited and inhibited by rewarding and aversive cues, respectively? OR are the arguing for discrete functional subclasses? The link between 'aversive stimuli' in the reference literature (often aversive tastants) and the link to the discussion of threat in the subsequent paragraph is not made explicit.

"To reveal functional VP neuron-types, we averaged first 1 s and last 5 s danger firing for each neuron, designating neurons with positive values as cue-excited (n = 131, ~51% of all cue-responsive neurons; Figure 2 top) and neurons with negative values as cue-inhibited (n = 126, ~49% of all cue-responsive neurons; Figure 2 bottom)." – The authors should clarify that they're looking at the difference between these values for determining cue-responsiveness, unless I'm mistaken, in which case they should clarify what they mean.

Statistical analyses seem rigorous, although they could use more description in places where noted. The authors state that baseline firing rate did not co-vary with cue responsiveness in ANCOVA, so then they subselect cue-inhibited neurons, cluster into high- intermediate- and low- firing within that subclass, and find that low-firing neurons showed greater danger inhibition than inhibition in response to uncertainty or safety? How do the authors distinguish between 'relative threat' and salience of the cue here?

Suppression ratio in Fig 1F, it should be shown how stable the behavior is across sessions; showing raw or average data from each subject across trials, rather than treating these observations as independent. The same is true of the statistics; is a nested or hierarchical design used to account for the lack of independence between observations? The authors should report bootstrap CI for differential suppression ratio for danger vs. safety in their initial behavioral results section.

Despite their subsequent analysis (Fig 4), the key observation that no class of neurons show firing differences that distinguish between 25% threat and safety seems to argue against the titular claim that VP neurons signal relative threat. Would the authors see a linear relationship with higher threat probabilities? The authors make a statement to this effect in their discussion: "Of course, differential cue firing would also be expected of a neural signal for fear output. Given that our rats showed

complete behavioral discrimination of danger, uncertainty and safety; inhibition of VP firing could reflect fear output, rather than relative threat.”

Standard errors should be shown on time courses of normalized firing rates (ie. Fig 3a,c,d, 5a, 6e).

Overall, the authors need to be much more clear about the impact of their findings, and how these results are incorporated into the larger body of work on the VP, and its known roles in salience and uncertainty processing. While the authors are correct that the VP is clearly heterogenous in terms of function, neurochemical and projection specificity, it's not clear how the clustering of units into low-medium- or high- firing advances our understanding of the VP.

Reviewer #3 (Remarks to the Author):

The authors demonstrate that two VP neuron populations, defined by firing property under single unit recordings, encode relative threat through decreases of firing. They trained rats to rewarded nose poke and fear discrimination task, which consists of three kinds of sound cues and associated foot-shock with unique probability. Single unit activities of VP neurons were recorded from 14 trained rats, and cue-inhibited and -excited neurons were analyzed separately. The authors grouped cue-inhibited neurons into three populations by baseline firing rates, and found that low and intermediate firing neurons convey relative threat by linear regression analysis. They also indicated that low firing neurons responded to both threat and reward in opposite way. Furthermore, they performed the same linear regression analysis on a single population of cue-excited neurons and revealed that cue-excited neurons signal both relative threat and fear output and increase firing to reward.

The methods sound solid, and the data are convincing. Although the manuscript is well-organized and well-written, there are some minor concerns that require further attention.

1) The authors used nose poke time as a measure of fear memory and the results of the suppression ratio are also clear. On the other hand, freezing behavior is also often used as an indicator of fear memory, and did the authors observe freezing behavior in the present study? If they recorded the behavior, it should be presented as supporting data. If not, this referee would like to see a more detailed explanation of why the authors used nose poke time in the present study.

2) When identifying cue-responsive neurons and determining their firing properties, the authors used "firing rate during the first 1 s and last 5 s of danger". A detailed explanation should be added for the first 1 s and last 5 s.

3) The authors use the k-means method to separate Low, Intermediate and High. Could you present the clusters visually? Fig S1 shows that there are some variations, and it seems that they were divided by the firing rate after all.

4) "measuring fear with conditioned suppression permitted us to record neural activity around reward delivery. Although not explicitly cued through the speaker, each reward delivery was preceded by a brief sound caused by the advance of the pellet feeder." (page 12)

It is not clear to this referee how exactly the unit recording for cue-reward was performed in this experiment.

5) In page 5, the authors use successive inequality signs like $(D > U \gg S)$ and $(D \gg U > S)$. The criteria for differences should be clearly stated.

6) In page 19, there is a reference number that should be superscript: “whereas DREADD inhibition has no impact⁵⁹.”

Reviewer #1 (Remarks to the Author):

Ventral pallidum is well known for its role in reward related behavior. Less is known about how this structure participates in threat. The authors used a conditioned suppression task while recording from VP neurons. VP neurons were highly sensitive to danger cues, with about half of neurons increase or decreasing. Baseline firing was not informative for cue increasing neurons but varied with decreasing neurons. Profiles of low, intermediate, and high baseline firing differed with respect to decreasing cue type. Low and intermediate decreased firing neurons signaled relative threat. Low firing decreasing neurons differently fired for danger cues and reward, others did not. Cue increasing neurons were scaled to danger and increased to reward, suggesting salience signaling. The experiments are well done with robust statistical analyses. My major concerns are with data interpretation.

Thank you for your time and feedback. We address each critique in turn below.

Major concerns:

1. Abstract writes an "integral role for the VP in threat-related behavior" but no test of this role was performed.

This is fair. We have revised the summary sentence of the abstract to better capture our findings: Abstract, last sentence, line 9: "The results reinforce anatomy to reveal the VP as a neural source of a dynamic, relative threat signal."

2. It was not clear how the authors could tell where their recording electrodes were located, at least in the picture shown in figure 1D. How can the authors determine track from recording location?

A small amount of current was passed through the wire tips during perfusions. This allowed us to visualize both the tip locations and the wire tracks in tissue sections. Starting with the electrode tips, we backwards calculated the driving path of the electrode through the brain. Only recording locations below the anterior commissure and inside of the substance P field were considered to be in the ventral pallidum. We have provided more details in Methods/Verifying electrode placement section, page 24, paragraph 1: "Passing current through the wire permitted the tip locations to be observed in brain sections. In addition, wire tracks leading up to tips were visible. Starting with the electrode tips, the driving path of the electrode through the brain was backwards calculated. Only recording locations below the anterior commissure and inside of the dense substance P field were considered to be in the VP (Fig. 1d). Single units recorded from all the recording sites within the boundaries of VP (Fig.1e) were included in analyses⁷⁴."

In addition, we have provided better markers of the track and recording site in Fig. 1d, page 5.

3. The analysis of single neurons and trials as subjects using t-tests violates the assumption of independence.

To address the concern for subjects, we have performed ANOVA and 95% bootstrap confidence intervals for mean individual suppression ratio. Both analyses reveal complete behavioral discrimination when individual data are examined (Fig. 1f, page 5).

We also understand the concern about treating single neurons as independent subjects in analyses. We should first note that this is the convention of the field. All VP single unit recording studies that we cite in this manuscript treated single units obtained from the same individual as independent observations. We would also argue that this convention is reasonable, as long as the single units are representative of the ventral pallidum. The best way to ensure representative recordings is to collect many ventral pallidum neurons from many individuals. We collected 435 units from 14 rats, a large sample size of neurons and subjects for single region recording study. One can then examine if neuron-types of interests were observed in many individuals – making those neuron-types more likely to be representative. We did this in our revised manuscript and found that Low and Intermediate firing neurons, but not High firing neurons, were likely to be representative of the ventral pallidum:

Page 7, paragraph 3, line 1: “Low firing neurons were observed in 11 of 14 individuals and Intermediate firing neurons in 9 of 14 individuals, making these neurons likely to be representative of the VP. High firing neurons were obtained in only 5 of 14 individuals, with 11 of 18 High firing neurons coming from a single individual (PA02, Supplementary Figure. 1). Because we cannot be certain High firing neurons are representative of the VP, primary analyses focus on Low and Intermediate firing neurons. High firing neuron analyses are provided as supplements (Supplementary Figure. 4, 5, 6a, d, and 7c).”

As a result, we have completely revised the manuscript to focus on Low and Intermediate firing neurons.

Another consideration is that even though some single units came from the same individual, those units were collected in sessions with a unique behavioral discrimination pattern. The session by session discrimination pattern can now be observed for each individual (Supplementary Figure 2, page 33). Even though overall discrimination is observed, individuals showed day-to-day variations in responding. This is particularly important, because our regression approach asks if single unit firing is described by behavior in that specific session. Two units signaling fear output that were recorded from the same individual, but in two different sessions, would be expected to show different patterns of responding if they signaled fear output.

4. It was not clear if reward related analyses were at the time of reward consumption or the click sound.

Activity was aligned to advancement of the pellet dispenser. The click sound is a byproduct of the dispenser advancing. We have clarified this description in the results:

Page 13, paragraph 3, line 1: “While our behavioral procedure is optimized to examine threat-related firing, ongoing reward seeking behavior permitted us to record neural activity around reward delivery. Although not explicitly cued through the speaker, each reward delivery was preceded by a brief sound caused by the advance of the pellet dispenser. Reward-related firing was extracted from inter-trial intervals, when no cues were presented. We asked if reward-related firing (time locked to pellet feeder advance) was observed in Low and Intermediate firing neurons.”

And methods:

Page 26, paragraph 1, line 10: “Although not explicitly cued through the speaker, each reward delivery was preceded by a brief sound caused by the advance of the pellet dispenser. For reward-related firing (time locked to pellet feeder advance), firing rate (Hz) was calculated in 250 ms bins from 2 s prior to 2 s following advancement of pellet dispenser, for a total of 16 bins. Mean differential firing was calculated for each of the 16 bins by subtracting pre-reward firing rate (mean of 1 s prior to reward delivery).”

5. The analysis on fear output signaling would be bolstered if video analyses confirmed freezing behavior related to the change in firing.

We do not have video for these recording sessions. However, we do not think that traditionally hand-scored video of freezing would provide an objective measure for our purposes. This is because our regression analysis requires precise levels of trial by trial fear to be measured. Even more, differential fear to danger, uncertainty and safety must be able to be detected. Studies that utilize hand-scoring of freezing typically do not examine discrimination procedures like ours, and instead must only determine overall fear levels to a fully predictive danger cue. Hand-scored freezing would also not specify the onset/offset of freezing bouts, which would be essential to determining VP firing reflecting freezing. Conditioned suppression provides an objective measure of fear that is precise at the trial level. In this manuscript, we show that cue-excited neuron firing is captured in part by fear output (via conditioned suppression). Previous studies from our

laboratory have found robust correlates of fear output – via conditioned suppression – in the ventrolateral periaqueductal gray (Wright and McDannald, 2019; Wright et al. 2019).

That said, we are aware that measuring freezing would be of value. Recently, the Witten lab has shown that freezing can be objectively measured at high temporal resolution using a convolutional neural network trained on hand-annotated data (Cai et al 2020 eLife, Distinct signals in medial and lateral VTA dopamine neurons modulate fear extinction at different times). We have obtained high speed video cameras and hardware/software to trigger video recording around cue presentation. We will soon be training a convolutional neural network to classify behavior in our discrimination procedure. So while the present study cannot benefit from this analysis that is very new to the field, future studies certainly will.

6. The discussion linking recorded cell-types to genetically defined neurons (interneurons, Penk, cholinergic, projections to specific targets, etc) is highly speculative without supporting data. Suggest remove much of this from the discussion.

As suggested, we have removed much of the cell-type speculation from the discussion.

Minor concerns:

1. Authors should be skeptical of literature claiming central amygdala projections to VP given that the extended amygdala system projects into the substantia innominata and other structures close to VP. Classic studies from Zahm, Heimer, and Alheid suggested the extended amygdala was largely distinct from the VP within the mesolimbic system.

We took the reviewers advice and went back to the cited papers to examine evidence of a central amygdala projection to the ventral pallidum. The following figure is a supplement from the Stephenson-Jones 2020 manuscript. Here, the authors examined brain regions providing direct input onto VP GABA and Glutamate neurons. Stephenson-Jones and colleagues report direct projections from the central amygdala to both VP GABA and Glutamate neurons (red boxes). An identical observation is made in the Tooley et al, 2018 paper we cite.

Figure S7. Monosynaptic inputs onto GABAergic and glutamatergic VP neurons. Related to Figure 7.

2. The sentence in the discussion regarding VP glutamate and GABA projections to amygdala - my knowledge of this subject, which may be limited, is that the cholinergic neurons project to amygdala primarily.

Cholinergic neurons comprise ~75% of the projections to the BLA with GABA comprising the remaining ~25%. We have edited the discussion to make sure this is clear:

Page 19, paragraph 2, line 7: “VP GABA neurons project to the BLA^{22,43,68}, comprising ~25% of the VP input to the BLA.”

Page 19, paragraph 2, line 12: “Low/Intermediate firing neurons may also include cholinergic neurons, which comprise ~75% of the VP input to the BLA^{70,71}, or may even include proenkephalin (Penk) neurons⁷².”

Reviewer #2 (Remarks to the Author):

This work by Moaddab and colleagues presents evidence from electrophysiology that VP neurons are poised to encode relative threat values. The study seems rigorous, the behavioral task elegant and the findings point to a role for VP neurons in signaling cues predicting aversive outcomes. There are a few points regarding statistical analysis (noted below) and graphical representation of results would benefit from clarification. The strength of the study would also be bolstered by contextualization of the findings, and a discussion of the novelty of the authors results compared to prior work that has established the VP as important for signaling aversive outcomes and punishment.

Thank you for the feedback. It was our intent to be rigorous! We address your concerns below.

The introduction does not well-contextualize the current state of the literature about the role of VP in reward and aversion, particularly given the known neurochemical heterogeneity that has been mapped on to functional heterogeneity (Knowland et al., Cell 2017, Faget et al., Nat Comm 2018, Tooley et al., Biol Psych 2018). Discussion of VP populations without mention of functional, anatomical or neurochemical specificity make it difficult to follow the logic of the introduction. Are the authors arguing that VP neurons are excited and inhibited by rewarding and aversive cues, respectively? OR are the arguing for discrete functional subclasses? The link between ‘aversive stimuli’ in the reference literature (often aversive tastants) and the link to the discussion of threat in the subsequent paragraph is not made explicit.

We have completely re-written the introduction to better contextualize the current literature, including discussion of neurochemical identity where appropriate.

“To reveal functional VP neuron-types, we averaged first 1 s and last 5 s danger firing for each neuron, designating neurons with positive values as cue-excited (n = 131, ~51% of all cue-responsive neurons; Figure 2 top) and neurons with negative values as cue-inhibited (n = 126, ~49% of all cue-responsive neurons; Figure 2 bottom).” – The authors should clarify that they’re looking at the difference between these values for determining cue-responsiveness, unless I’m mistaken, in which case they should clarify what they mean.

Thank you for catching this. Values were compared to zero and neurons with positive values deemed cue-excited while neurons with negative values deemed cue-inhibited. We have revised this section to clarify.

Page 5, paragraph 2, line 3: “To reveal functional VP neuron-types, we averaged the first 1 s and last 5 s danger firing rates for each neuron to obtain a single value and compared this value to zero. Neurons with positive values (>0) increased danger firing rate over baseline and were designated as cue-excited (n = 131, ~51% of all cue-responsive neurons; Fig. 2 top). Neurons with negative values (<0) decreased firing rate below baseline and were designated as cue-inhibited (n = 126, ~49% of all cue-responsive neurons; Fig. 2 bottom).”

Statistical analyses seem rigorous, although they could use more description in places where noted. The authors state that baseline firing rate did not co-vary with cue responsiveness in ANCOVA, so then they subselect cue-inhibited neurons, cluster into high- intermediate- and low- firing within that subclass, and find that low-firing neurons showed greater danger inhibition than inhibition in response to uncertainty or safety? How do the authors distinguish between ‘relative threat’ and salience of the cue here?

We apologize for the confusion. ANCOVA was separately performed for cue-excited and cue-inhibited neurons. ANCOVA for cue-excited neurons found that cue firing did not covary with baseline firing rate. For this reason, all cue-excited neurons were analyzed together.

ANCOVA for cue-inhibited neurons found that cue firing covaried with baseline firing rate. This is why we only applied k-means clustering to cue-inhibited neurons. Importantly, k-means clustering can identify different functional types, but cannot provide insight into what is signaled by each type. Signals for relative threat and salience can only be determined by examining patterns of cue firing, then regression.

Suppression ratio in Fig 1F, it should be shown how stable the behavior is across sessions; showing raw or average data from each subject across trials, rather than treating these observations as independent. The same is true of the statistics; is a nested or hierarchical design used to account for the lack of independence between observations? The authors should report bootstrap CI for differential suppression ratio for danger vs. safety in their initial behavioral results section.

We have made supplemental figures showing mean behavior for each individual for all recording sessions with cue-responsive neurons (Supplementary Figure 1, page 32) as well as session x session behavior for each individual (Supplementary Figure 2, page 33). Behavior varied across sessions but was relatively stable for each individual. The most consistent trend was for discrimination to improve over the course of recording. This is expected, given that surgery + recovery would mean the first recording session was at least 13 days since the last pretraining behavior session.

We do not use a nested or hierarchical design for behavior analysis. We now present ANOVA results for mean suppression ratio for each individual, meaning that each individual now contributes only one data point per cue. We still find complete discrimination: ANOVA main effect of cue. 95% bootstrap CIs are now calculated for all comparisons and each CI does not contain zero.

Page 4, last paragraph, line 3: “Suppression ratios were high to danger, intermediate to uncertainty, and low to safety (Fig.1f). Analysis of variance (ANOVA) for mean individual suppression ratio [factor: cue (danger, uncertainty, and safety)] revealed a main effect of cue ($F_{2,26} = 75.34$, $p=1.52 \times 10^{-11}$, partial eta squared (η_p^2) = 0.85, observed power (op) = 1.00). Differential suppression ratios were observed for each cue pair. The 95% bootstrap confidence interval for differential suppression ratio did not contain zero for danger vs. uncertainty (mean = 0.28, 95% CI [(lower bound) 0.19, (upper bound) 0.38]), uncertainty vs. safety (M = 0.52, 95% CI [0.35, 0.65]), and danger vs. safety (M = 0.80, 95% CI [0.65, 0.98]; Fig.1f)”

Despite their subsequent analysis (Fig 4), the key observation that no class of neurons show firing differences that distinguish between 25% threat and safety seems to argue against the titular claim that VP neurons signal relative threat. Would the authors see a linear relationship with higher threat probabilities? The authors make a statement to this effect in their discussion: “Of course, differential cue firing would also be expected of a neural signal for fear output. Given that our rats showed complete behavioral discrimination of danger, uncertainty and safety; inhibition of VP firing could reflect fear output, rather than relative threat.”

This is an insightful observation. This made us wonder if the pattern of differential cue firing in Low and Intermediate neurons was best described by the actual foot shock probability associated with each cue: 0.00, 0.25 and 1.00 or perhaps alternative probabilities? The reviewer rightly points out that neurons may fire identically to safety and uncertainty, in which case their uncertainty firing would better be captured by $p=0.00$.

To address this, we performed an entirely new regression analyses in which we systematically varied the relative threat regressor. The values assigned to danger (1.00) and safety (0.00) were fixed, but the value assigned to uncertainty was incremented from 0 to 1 in 0.25 steps (0.00, 0.25, 0.50, 0.75, and 1.00). We performed separate regression for each uncertainty assignment, obtaining relative threat beta coefficients for each analysis. We then performed ANOVA, comparing relative threat beta coefficients for uncertainty assignments. The comparisons of greatest interest were those for 0.25 (actual probability) vs. 0.00 (probability equating uncertainty to safety); and 0.25 vs. 0.50 (midpoint probability, exceeding actual probability).

For both Low and Intermediate firing neurons, the actual probability (0.25) better captured the pattern of differential cue firing than did the probability equating uncertainty to safety (0.00). This made us confident that the neurons are treating the uncertainty cue differently than the safety cue. Surprising to us, Low and Intermediate firing neurons were similarly described by uncertainty assignments of 0.25 and 0.50. Which led us to this new analysis:

Page 11, last paragraph, line 1: “We were curious whether relative threat signaling of the actual probability was indistinguishable from the midpoint probability, or whether signaling dynamically changed as foot shock drew near. Now, we performed single unit regression using relative threat regressors with uncertainty assignments of 0.25 and 0.50. The resulting beta coefficients were subjected to ANOVA [factors: assignment (0.25

and 0.50), and interval (1 s bins from 2 s prior to cue onset → 2 s following cue offset)]. Low firing neurons initially decreased firing according to the actual shock probability (0.25), but later decreased firing according to the greater-than-actual, midpoint probability (0.50) (Fig. 4b). In support, ANOVA found an assignment x interval interaction ($F_{13, 949} = 2.42$, $p=0.003$, $\eta_p^2 = 0.03$, $op = 0.98$). Confirming initial signaling of the actual shock probability, early beta coefficients (first 4 s of cue) were shifted below zero for the 0.25 uncertainty assignment ($M = -0.57$, 95% CI [-1.02, -0.10]), but not for the 0.50 uncertainty assignment ($M = 0.06$, 95% CI [-0.38, 0.48]; Fig. 4c, left). Confirming late signaling of the midpoint probability, late beta coefficients (last 2 s cue plus 2 s delay) were shifted below zero for the 0.5 uncertainty assignment ($M = -0.51$, 95% CI [-1.03, -0.03]), but not for the 0.25 uncertainty assignment ($M = 0.20$, 95% CI [-0.22, 0.62]; Fig. 4c, right). Consistent with the ANOVA interaction, there was a positive early-to-late shift in beta coefficients for the 0.25 uncertainty assignment ($M = 0.76$, 95% CI [0.41, 1.09]), but a negative shift for the 0.50 uncertainty assignment 0.50 ($M = -0.56$, 95% CI [-0.95, -0.16]), and these shifts differed from one another ($M = 1.32$, 95% CI [0.57, 2.04]; Fig. 4d).”

In short, the new analysis supports our claim that VP neurons signal relative threat, but do so dynamically. At cue onset, VP neurons linearly decrease firing according to shock probability. This is a fairly pure signal for relative threat. As the cue continues, VP neurons show disproportionate firing decreases to uncertainty.

Standard errors should be shown on time courses of normalized firing rates (ie. Fig 3a, c, d, 5a, 6e).

Standard error bars have been added to all normalized firing rate figures. (see Fig. 3 a, c; Fig. 5a; Fig. 6 a, e; Supplementary Figure. 4 a, c; Supplementary Figure. 6 a; Supplementary Figure. 9 a)

Overall, the authors need to be much more clear about the impact of their findings, and how these results are incorporated into the larger body of work on the VP, and its known roles in salience and uncertainty processing. While the authors are correct that the VP is clearly heterogenous in terms of function, neurochemical and projection specificity, it's not clear how the clustering of units into low- medium- or high- firing advances our understanding of the VP.

We have completely revised our discussion to make clear the advance of the current study, as well as how the VP neuron-types we observed may map onto known VP neuron types with respect to neurochemical identity.

Reviewer #3 (Remarks to the Author):

The authors demonstrate that two VP neuron populations, defined by firing property under single unit recordings, encode relative threat through decreases of firing. They trained rats to rewarded nose poke and fear discrimination task, which consists of three kinds of sound cues and associated foot-shock with unique probability. Single unit activities of VP neurons were recorded from 14 trained rats, and cue-inhibited and -excited neurons were analyzed separately. The authors grouped cue-inhibited neurons into three populations by baseline firing rates, and found that low and intermediate firing neurons convey relative threat by linear regression analysis. They also indicated that low firing neurons responded to both threat and reward in opposite way. Furthermore, they performed the same linear regression analysis on a single population of cue-excited neurons and revealed that cue-excited neurons signal both relative threat and fear output and increase firing to reward. The methods sound solid, and the data are convincing. Although the manuscript is well-organized and well-written, there are some minor concerns that require further attention.

Thank you for your feedback. We spent a fair amount of time analyzing these data and writing this manuscript; it is satisfying to hear that it reads well. We address your concerns below.

1) The authors used nose poke time as a measure of fear memory and the results of the suppression ratio are also clear. On the other hand, freezing behavior is also often used as an indicator of fear memory, and did the authors observe freezing behavior in the present study? If they recorded the behavior, it should be presented as supporting data. If not, this referee would like to see a more detailed explanation of why the authors used nose poke time in the present study.

Reviewer 2 raised a similar point. We do not have video for these recording sessions. However, we do not think that traditionally hand-scored video of freezing would provide an objective measure of fear for our purposes. This is because our regression analysis requires precise levels of trial by trial fear to be measured. Even more, differential fear to danger, uncertainty and safety must be able to be detected. Studies that utilize hand-scoring of freezing typically do not examine discrimination procedures like ours, and instead must only determine overall fear levels to a fully predictive danger cue. Hand scored freezing would also not specify the onset/offset of freezing bouts, which would be essential to determining VP firing reflecting freezing. Conditioned suppression provides an objective measure of fear that is precise at the trial level. In the manuscript we show that cue-excited neuron firing is captured in part by fear output (via conditioned suppression). Previous studies from our laboratory have found robust correlates of fear output – via conditioned suppression – in the ventrolateral periaqueductal gray (Wright and McDannald, 2019; Wright et al. 2019).

That said, we aware that measuring freezing would be of value. Recently, the Witten lab has shown that freezing can be objectively measured at high temporal resolution using a convolutional neural network trained on hand-annotated data (Cai et al 2020 eLife, Distinct signals in medial and lateral VTA dopamine neurons modulate fear extinction at

different times). We have obtained high speed video cameras and hardware/software to trigger video recording around cue presentation. We will soon be training a convolutional neural network to classify behavior in our discrimination procedure. So while the present study cannot benefit from this analysis that is very new to the field, future studies certainly will.

2) When identifying cue-responsive neurons and determining their firing properties, the authors used "firing rate during the first 1 s and last 5 s of danger". A detailed explanation should be added for the first 1 s and last 5 s.

The text has been revised as suggested. The revised text reads as follows:

Page 4, Paragraph 2, line 3; "To identify cue-responsive neurons in an unbiased manner, we compared mean firing rate (Hz) during the 10 s prior to cue presentation (baseline), to mean firing rate (Hz) during the first 1 s and last 5 s of cue presentation. A neuron was considered cue-responsive if it showed a significant change in firing from baseline (increase or decrease; paired, two-tailed t-test, $p < 0.05$) to danger, uncertainty or safety during the first 1 s or the last 5 s interval".

Methods, identifying cue-responsive units section, page 24, paragraph 4, last sentence; "Single units were screened for cue responsiveness by comparing mean firing rate (Hz) during the 10 s prior to cue presentation (baseline), to mean firing rate (Hz) during the first 1 s and last 5 s of cue presentation. A neuron was considered cue-responsive if it showed a significant change in firing from baseline (increase or decrease; paired, two-tailed t-test, $p < 0.05$) to danger, uncertainty or safety during the first 1 s or the last 5 s interval".

3) The authors use the k-means method to separate Low, Intermediate and High. Could you present the clusters visually? Fig S1 shows that there are some variations, and it seems that they were divided by the firing rate after all.

Supplementary Figure 3 (page 34) now shows the distribution of single units for each cluster, for each characteristic: coefficient of variance, coefficient of skewness, waveform half duration, waveform amplitude and baseline firing rate.

The reviewer is also correct that the neurons were *mostly* divided by firing rate. However, ANOVA found that coefficient of variance, coefficient of skewness, and waveform half duration also contributed to the final clustering result. These results are now provided:

Page 7, paragraph 2, line 3: "To identify distinct functional neuron-types, we used k-means clustering for baseline firing and four additional characteristics: coefficient of variance^{52,53}, coefficient of skewness⁵³, waveform half duration⁵⁴, and waveform amplitude ratio⁵⁴ (see methods for full description of each). ANOVA revealed four of the five characteristics significantly contributed to clustering, with baseline firing rate being the greatest contributor (baseline, $F_{2,123} = 546.73$, $p = 6.25 \times 10^{-62}$; coefficient of variance, $F_{2,123} = 8.79$, $p = 0.0003$; coefficient of skewness, $F_{2,123} = 18.20$, $p = 1.20 \times 10^{-7}$; half duration,

$F_{2,123} = 17.90$, $p=1.50 \times 10^{-7}$; and amplitude ratio, $F_{2,123} = 2.12$, $p=0.12$; firing and waveform characteristics can be found in Supplementary Figure. 3)."

4) "measuring fear with conditioned suppression permitted us to record neural activity around reward delivery. Although not explicitly cued through the speaker, each reward delivery was preceded by a brief sound caused by the advance of the pellet feeder." (page 12)

It is not clear to this referee how exactly the unit recording for cue-reward was performed in this experiment.

We agree and have updated the text.

Page 13, last paragraph, line 2; "Although not explicitly cued through the speaker, each reward delivery was preceded by a brief sound caused by the advance of the pellet dispenser. Reward-related firing was extracted from inter-trial intervals, when no cues were presented. We asked if reward-related firing (time locked to pellet feeder advance) was observed in Low and Intermediate firing neurons".

5) In page 5, the authors use successive inequality signs like $(D > U \gg S)$ and $(D \ggg U > S)$. The criteria for differences should be clearly stated.

Thank you for pointing this out. We have removed the signs and updated the text.

Page 10, paragraph 2, line 2; "Fear output and relative threat could be dissociated because rats showed greater fear to uncertainty than would be expected based on its foot shock probability (Fig. 1f)".

6) In page 19, there is a reference number that should be superscript: "whereas DREADD inhibition has no impact⁵⁹."

As suggested by the reviewer, the correction has been made.

REVIEWERS' COMMENTS:

Reviewer #1 (Remarks to the Author):

The authors have satisfied my prior concerns

Reviewer #2 (Remarks to the Author):

I very much enjoyed reading the revised version of this manuscript. In this paper, Moaddab and colleagues provide evidence for relative threat discrimination in the ventral pallidum. Their analysis is rigorous, and any ambiguities regarding the interpretation of results raised in the last round of revisions have been carefully and thoroughly addressed with additional details in the text as well as completely new analyses. Moreover, concerns I had about contextualizing these novel results (pertaining specifically to the role of the VP in encoding relative threat) within the literature have been addressed by substantial re-writes to the introduction and discussion.

I particularly appreciate and commend the authors' efforts to balance the often competing demands of reviewers in their revision (for example, elaborating on the discussion of neurochemical heterogeneity within the VP vs. suggestions of a different reviewer to eliminate this altogether, based on its speculative nature). Overall, I think the manuscript will be an important addition to the field, as it extends our understanding of the role of the ventral pallidum in encoding relative reward, threat, punishment; understanding the neural basis for encoding and integration of these signals will be critical for understanding how the brain mounts adaptive responses to threatening stimuli. I am happy to endorse this publication.

Reviewer #3 (Remarks to the Author):

All the points are clearly explained and properly revised in the present manuscript. This reviewer is satisfied with the present version for publication.